# Intraobserver and interobserver agreement among anterior chamber angle evaluations using automated 360-degree gonio-photos

**Masato Matsuo**[1]*, **Shiro Mizoue**[2], **Koji Nitta**[3], **Yasuyuki Takai**[1], **Kazunobu Sugihara**[1], **Masaki Tanito**[1]

1 Department of Ophthalmology, Shimane University Faculty of Medicine, Izumo, Japan, 2 Department of Ophthalmology, Ehime University Graduate School of Medicine, Matsuyama, Japan, 3 Department of Ophthalmology, Fukui-ken Saiseikai Hospital, Fukui, Japan

* mmpeaceful@yahoo.ne.jp

**Data Availability Statement:** All relevant data are within the manuscript and its Supporting information files.

## Abstract

### Purpose

To investigate the reproducibility for the iridocorneal angle evaluations using the pictures obtained by a gonioscopic camera, Gonioscope GS-1 (Nidek Co., Gamagori, Japan).

### Methods

The pragmatic within-patient comparative diagnostic evaluations for 140 GS-1 gonio-images obtained from 35 eyes of 35 patients at four ocular sectors (superior, temporal, inferior, and nasal angles) were conducted by five independent ophthalmologists including three glaucoma specialists in a masked fashion twice, 1 week apart. We undertook the observer agreement and correlation analyses of Scheie's angle width and pigmentation gradings and detection of peripheral anterior synechia and Sampaolesi line.

### Results

The respective Fleiss' kappa values for the four elements between manual gonioscopy and automated gonioscope by the glaucoma specialist were 0.22, 0.40, 0.32 and 0.58. Additionally, the respective intraobserver agreements for the four elements by the glaucoma specialist each were 0.32 to 0.65, 0.24 to 0.71, 0.35 to 0.70, and 0.20 to 0.76; the Fleiss' kappa coefficients for the four elements among the three glaucoma specialists were, respectively, 0.31, 0.38, 0.31, and 0.17; the Fleiss' kappa coefficients for the angle width and pigmentation gradings between the two glaucoma specialists each were 0.30 to 0.35, and 0.29 to 0.43, respectively. Overall, the Kendall's tau coefficients for the angle gradings reflected the positive correlations in the evaluations.

### Conclusion

Our findings suggested slight-to-substantial intraobserver agreement and slight-to-fair (among the three) or fair-to-moderate (between the two each) interobserver agreement for the angle assessments using GS-1 gonio-photos even by glaucoma specialists. Sufficient

**Funding:** The authors received no specific funding for this work.

**Competing interests:** The authors have declared that no competing interests exist.

training and a solid consensus should allow us to perform more reliable angle assessments using gonio-photos with high reproducibility.

## Introduction

Glaucoma is the leading cause of irreversible blindness worldwide, with about 8.4 million people blind from the disease, and iridocorneal angle evaluation by gonioscopy is necessary for glaucoma diagnosis and clinical evaluation [1–3]. Elevated intraocular pressure (IOP) is the only modifiable and independent risk factor for development and progression of this optic neuropathy. The mechanism of IOP elevation depends on the anatomic and structural status of the drainage system, or anterior chamber angle, and separates the pathology into two major subtypes: open-angle and closed-angle glaucoma.

Primary open-angle glaucoma (POAG) is the most common type, with an estimated 45 million people worldwide with open-angle glaucoma (OAG). This condition is associated with an open anterior chamber angle without other known explanations (i.e., secondary glaucoma) by gonioscopy for progressive glaucomatous optic nerve change [1]. OAG includes pseudoexfoliation glaucoma, which is considered to be the most common type of secondary glaucoma and can advance rapidly with continuous high IOP and be refractory to several therapeutic interventions [4]. The high pigmentation in trabecular meshwork (TM) and Sampaolesi's line are important for the diagnosis. In OAG, the IOP is regulated primarily by resistance to aqueous humor outflow through the TM [5]. High pigmentation levels in the TM may contribute to increased outflow resistance and increased IOP. Therefore, assessment of the TM pigmentation is necessary for glaucoma diagnosis and clinical evaluation [3, 6]. Moreover, the other types of secondary glaucoma also require the examination for angle chromatic information to identify underlying causes that may alter the treatment plan.

While primary angle-closure glaucoma (PACG), which is characterized by elevated IOP as a result of mechanical obstruction of the TM by either apposition of the peripheral iris to the TM or an angle closed by synechia, causes more blindness than POAG, particularly in Asia. According to the International Society of Geographic and Epidemiologic Ophthalmology (ISGEO) protocol, primary angle closure diseases were classified into 3 categories: primary angle closure suspect (PACS), primary angle closure (PAC) and PACG, which means a continuum between open and closed angles. Thus, early detection of a narrow angle and peripheral anterior synechia (PAS) and appropriate treatment in earlier stages by laser iridotomy or lens extraction can prevent the progression and the glaucomatous optic neuropathy [7, 8]. Therefore, gonioscopy is essential for glaucoma diagnoses and clinical evaluations.

The clinical standard for evaluating the TM remains gonioscopy [3, 9, 10], because no other method is useful for estimating the chromatic information of the TM. Although alternative methods for evaluating the iridocorneal angle, such as anterior-segment optical coherence tomography (AS-OCT) and ultrasound biomicroscopy (UBM), are commonly used, these techniques only provide anatomic quantification. Alternatively, conventional gonioscopy requires skill and time, and it is difficult to obtain images of sufficient quality in a standardized manner because of the manual requirements. Moreover, gonioscopy can examine at one time only a limited contiguous portion of the iridocorneal angle, and the technique relies on subjective assessment; therefore, it is problematic to assess gonioscopic findings because of its substantial intraobserver and interobserver variability [11, 12]. Despite its importance, clinicians still do not use gonioscopy enough, with usage rates during first-time glaucoma clinic visits

ranging from 17.96% to 45.9%. Moreover, a low rate of gonioscopy (49%) also was reported during the 4 to 5 years preceding glaucoma surgery [13].

The Gonioscope GS-1 (Nidek Co., Gamagori, Japan) is a recently released gonioscopic camera that covers 360 degrees of the angle and provides true-color gonio-images automatically in a standardized manner in less than 1 minute/eye. As these images can be analyzed post-hoc, physicians can make detailed observations, including image manipulation to magnify any abnormalities. This technology is intended to assist in decision-making concerning the degree of angle opening or suitability for performing certain procedures, such as laser trabeculoplasty or angle-based surgical procedures. Moreover, the technology should enable telemedicine or tele-glaucoma care with standardized gonio-photos by glaucoma specialists in remote locations, which can contribute to improved glaucoma diagnostic rates and reduce preventable visual loss if the disease is detected early enough [14]. However, as with conventional gonioscopy, there likely is observer variability associated with assessments using the GS-1 gonio-photos, which should be evaluated in detail. Teixeira et al. first reported interobserver agreement for Shaffer's grading, angle closure, and detection of other angle structural abnormalities using the prototype GS-1 gonio-photos obtained by an ophthalmology resident and a glaucoma specialist [9]; the intraobserver and interobserver agreements of angle evaluations by ophthalmologists, especially glaucoma specialists, remain unknown. Moreover, the angle pigmentation grades were not assessed in the study, and angle evaluations were performed only inferonasally, inferotemporally, superotemporally, and superonasally, which are not common gonioscopic assessment sites and recordings. Ideally, this validation should be performed in a diagnostic agreement study by well-trained ophthalmologists familiar with gonioscopy and among various types of patients that can potentially demonstrate that the images collected through the device will allow the same diagnosis by different doctors. In the current study, we investigated the observer agreements of independent ophthalmologists including glaucoma specialists in multiple centers for angle evaluations in four primary sectors, inferiorly, superiorly, temporally, and nasally, using the images obtained by a new commercial version of the Gonioscope GS-1.

## Methods

The institutional review boards of Shimane University Faculty of Medicine, Izumo, Japan, Ehime University Graduate School of Medicine, Matsuyama, Japan and Fukui-ken Saiseikai Hospital, Fukui, Japan, reviewed and approved the research, respectively, and waived written informed consent. Each participating center has a specialized glaucoma unit. All research adhered to the tenets of the Declaration of Helsinki.

### Overview of the study design

The pragmatic within-patient comparative diagnostic evaluations for 140 GS-1 gonio-images by five independent ophthalmologists (S.M., K.N., Y.T., K.S., and M.T.) were conducted in a masked fashion twice 1 week apart. Three of the five ophthalmologists were glaucoma specialists (S.M., K.N., and M.T.) belonging to different tertiary care centers. We evaluated the intraobserver and interobserver agreement values of the Scheie's angle width and pigmentation gradings and detection of PAS and the Sampaolesi line. In the Scheie's grading system, the angle widths are defined as wide if all structures are visible up to the iris root and its attachment to the anterior ciliary body; grade I if all angle structures are visible up to the scleral spur; grade II when angle structures are visible only up to the posterior TM; grade III when only the Schwalbe lines and the anterior TM are visible; and grade IV when no TM is observed. We adopted the grading system for the experimental design. Moreover, angle pigmentation was

graded from 0 (no pigmentation) to IV (severe pigmentation) with increasing degrees of pigmentation by the Scheie grading system [15]. The Department of Ophthalmology, Shimane University Faculty of Medicine, was responsible for the test data preparation, data acquisition, and analyses.

## Study subjects

Study subjects were identified from the medical records of patients who underwent successful evaluations using the Gonioscope GS-1 images obtained in four sectors, superiorly, temporally, inferiorly, and nasally, at Shimane University Hospital from October 2018 to January 2019. Each patient underwent a comprehensive eye examination that included autorefraction (RC-5000, Tomey, Nagoya, Japan), visual acuity measurement, slit-lamp examination, IOP measurement using Goldmann applanation tonometry (Haag-Streit, Koniz, Switzerland), conventional static and dynamic gonioscopy with an Ocular Magna View Two-Mirror Gonio (Ocular Instruments, Bellevue, WA, USA) under slit-lamp illumination, corneal thickness measurement by specular microscopy (EM-3000; Tomey), central anterior chamber depth measurement using OA-2000 (Tomey), fundus examination, and other examinations as determined by the clinician as part of the clinical examination. Manual gonioscopy was performed by a single glaucoma specialist (M.T.) and angle width grades were evaluated with the modified Shaffer's grading system for the clinical assessment. In the Shaffer scheme, grade 0 was considered for the 0° wide angles when no TM could be observed; grade 1 was recorded for the 5° to 15° wide angles when only Schwalbe's lines and the anterior TM were visible; grade 2 was assigned for 15° to 25° wide angles when angle structures were visible only until the posterior TM; grade 3 was used for 25° to 35° wide angles when all angle structures were visible up to the scleral spur, and grade 4 for greater than 35° wide angles if all structures were visible up to the iris root and its attachment to the anterior ciliary body [9, 16]. The patient age, gender, clinical history, ocular characteristics, and clinical diagnosis of each study eye were recorded.

Ultimately, the study included gonio-images obtained in four ocular sectors of 35 eyes of 35 Japanese patients: five eyes of five normal patients, five eyes with ocular hypertension, 10 eyes with POAG, five eyes with pseudoexfoliation glaucoma, five eyes with secondary OAG, and five eyes with PACG. Glaucomatous eyes were defined based on the characteristic optic disc appearance (localized or diffuse neuroretinal rim thinning/notching), the presence of a retinal nerve fiber layer (RNFL) defect in the corresponding region, and the presence of a visual field defect corresponding to the structural change with Humphrey Field Analyzer (Carl Zeiss, Dublin, CA, USA). The glaucoma types were diagnosed via slit-lamp examination and manual gonioscopy. The normal controls had IOPs less than or equal to 21 mmHg, no history of IOP elevation, no glaucomatous optic disc appearance, and no RNFL defect; eyes with ocular hypertension had IOPs over 21 mmHg, no secondary cause of IOP elevations, no glaucomatous optic disc appearance, and no RNFL defect [17].

Subsequently, one masked observer (M.M.) assessed the image quality of the gonio-images based on previous studies (i.e., grade 0 indicated clear and focused; grade 1 slightly blurred with visible details; and grade 2 blurred with no discernible details) [9, 16]. The study excluded whole eyes with poor-quality images that had at least one grade 2 image in four ocular sectors. After the image quality assessments, 17 poor-quality ones (23.9%) were excluded from angle evaluations, which were composed of 11 eyes of 11 subjects for one eye and 6 eyes of 3 subjects for both eyes. If both eyes met the eligibility criteria, the eye with the better-quality images was selected [3].

The other exclusion criteria were a history of trauma, intraocular surgery including uncomplicated cataract surgery, laser iridotomy, and laser trabeculoplasty that could potentially

change the intra-anterior chamber environment and drastically alter angle structures and TM pigmentation [3].

## Gonioscope GS-1 imaging

The Gonioscope GS-1 system includes a 16-mirror faceted automatically rotating optical contact prism illuminated by a white light-emitting diode lamp, and a built-in, high-resolution color camera. Each facet of the prism projects white light to a 22.5-degree portion of the angle. The gonioscope camera can take 17 pictures simulating indirect static gonioscopy at varying focal depths from each of the facets, for a total of 272 gonio-photos according to the same protocol as defined by the manufacturer [3, 9, 13].

After instilling topical anesthetic eye drops of 0.4% oxybuprocaine hydrochloride (Benoxil ophthalmic solution 0.4%, Santen, Osaka, Japan) into the eye and using a lens coupling gel (GenTeal Gel, Alcon, Fort Worth, TX, USA), the images were captured by the Gonioscope GS-1 with the participants seated in a darkened room and looking in primary gaze. During the first step of the image-acquisition process, the instrument was moved manually toward the apex of the patient's cornea until light contact was obtained and a focused image appeared on the gonioscope screen. Unlike manual gonioscopy, the proprietary pressure sensor feedback mechanism built into the GS-1 allows for sufficient contact between the cornea and gonioprism coupled with a gel ointment for imaging. A fixation target in the center of the gonio-prism stabilized the ocular movements while the prism touched the eye surface; if any indentation was detected, the device did not proceed, and no images were acquired until the indention was eliminated. Second, the machine automatically achieves fine focus on the angle structures and takes 16 sequential high-resolution photographs at multiple focal planes. With additional manual focusing time, examining one eye requires less than 1 minute [3, 9, 13].

## Evaluation of GS-1 gonio-images

**Interobserver agreement and correlation analyses using GS-1 gonio-photos.** The five independent ophthalmologists were board-certified ophthalmic specialists of the Japanese Ophthalmological Society (JOS) who had completed 5 years of training at a JOS-approved educational institution of ophthalmology and passed an examination administered by the JOS. The three glaucoma specialists in three different tertiary care centers diagnosed and treated glaucoma patients daily. One of the authors (M.M.) provided the participating ophthalmologists with a short explanation about angle evaluation with GS-1 gonio-images regarding the 140 randomized GS-1 gonio-images (first test, S1 Data) and the answer sheets were sent by e-mail. Thereafter, the five ophthalmologists including the three glaucoma specialists performed the diagnostic evaluations during the first test without specific clinical information about the patients. The Shaffer's grading system is commonly used in daily clinical practice; however, it depends on both the visibility of the anatomical structures of the angle and its angularity. While the Scheie's angle width grading system is also commonly used gonioscopic grading scheme based upon the visibility of the anatomical structures of the angle only, therefore, we thought the Scheie's system was more appropriate for angle evaluations with GS-1 gonio-photos, because it is easy to judge the visibility of the anatomical structures but it is difficult to expect the angularity by one planar image. Thus, each ophthalmologist was asked to individually evaluate the width of Scheie's angle and pigmentation gradings [15] and the presence or absence of PAS and Sampaolesi line [9]. For analysis, we modified the original Scheie's grading systems by labeling 0 (wide or 0), 1 (I), 2 (II), 3 (III), and 4 (IV) [18]. Then, we analyzed the interobserver agreements for angle evaluations using the gonioscopic photos.

**Intraobserver reproducibility for angle evaluations between manual gonioscopy and automated gonioscope.** To evaluate the intraobserver reproducibility for angle evaluations, we compared the outcomes with manual gonioscopy and those with automated gonioscope by the glaucoma specialist (M.T.) in the first test. For analysis, the Shaffer's angle width grade with gonioscopy were translated to the Scheie's angle width grade by converting wide or 0 (4), I (3), II (2), III (1), and IV (0). We also analyzed the intraobserver reproducibility for closed angle detection in each sector (n = 140) and in each eye (n = 35). Additionally, for analysis, we defined the "closed angle sector" as grade 0 or 1 in the Shaffer's system with gonioscopy and regarded it as grade III or IV in the Scheie's system with GS-1 image evaluation, and we also defined the "closed angle eye" when 3 or 4 sectors in the eye were closed.

**Intraobserver agreement and correlation analyses using GS-1 gonio-photos.** One week after the first test, the five ophthalmologists received the different randomized GS-1 gonio-images from the same set (second test, S2 Data) and the answer sheet and conducted the diagnostic evaluations during the second test without specific clinical information about the patients. This was performed to determine the intraobserver agreements for angle evaluations using the gonioscopic photos.

## Statistical analysis

We evaluated the agreement levels of the evaluations of Scheie's angle width and pigmentation gradings and each of the possible gonioscopic findings in the detection of PAS and the Sampaolesi line [9, 18]. Fleiss' kappa statistic was used to assess the agreement for binary and nominal variables, including the intraobserver and interobserver agreement [19]. A bootstrap method was used to calculate the 95% confidence intervals [20]. The analyses were performed in R statistical software version 3.5.3 [21]. The agreement of the kappa statistics was interpreted as poor (~0.20), fair (0.20–0.40), moderate (0.40–0.60), substantial (0.60–0.80), and excellent (0.80~) based on the proposal by Altman [22, 23]. Kendall rank correlation coefficients were also calculated to assess the statistical associations based on the ranks of the data by using JMP Pro 14 software (SAS Institute Japan Inc., Tokyo, Japan). The clinical and demographic characteristics are expressed as the mean and standard deviation for continuous variables or by number and frequency for discrete variables.

## Results

A total of 140 images of 35 eyes of 35 subjects were included in the study. Table 1 shows the demographic data and clinical characteristics of the study subjects. The participants were Japanese (mean age, 61.5 ± 14.3 years; range, 23–83). Women (n = 14) comprised 40% of the subjects.

## Intraobserver reproducibility for angle evaluations between manual gonioscopy and automated gonioscope

Table 2 shows the agreement analyses for angle evaluations between manual gonioscopy and automated GS-1 gonioscope by the single glaucoma specialist (M.T.). One hundred forty positions from the four ocular sectors of 35 eyes were evaluated. Compared with the results of gonioscopy, the outcomes of Scheie's angle pigmentation grading, PAS detection and the Sampaolesi line detection using GS-1 gonio-photos were significantly overestimated (P<0.05, respectively), and the outcome of Scheie's angle width grading by automated gonio-photos also tended to be overestimated. The Fleiss' kappa values for the four elements, i.e., Scheie's angle width and pigmentation gradings and for detection of PAS and Sampaolesi line were 0.22 (fair), 0.40 (fair), 0.32 (fair) and 0.58 (substantial), respectively. In addition, the Kendall's

**Table 1. Demographics and clinical characteristics of the study subjects.**

| Characteristics | Total (n = 35) |
|---|---|
| Age, years, mean ± SD (range) | 61.5±14.3 (23–83) |
| Women, number (%) | 14 (40) |
| Right eye, number (%) | 19 (54) |
| LogMAR best corrected visual acuity, mean ± SD (range) | 0.11±0.37 (-0.08–1.85) |
| Intraocular pressure, mmHg, mean ± SD (range) | 18.3±8.5 (9–57) |
| Corneal curvature, diopters, mean ± SD (range) | 43.5±1.6 (40.4–47.9) |
| Spherical equivalent refractive error, diopters, mean ± SD (range) | -2.4±3.3 (-10.8–1.6) |
| Corneal thickness, μm, mean ± SD (range) | 525.5±32.5 (437–585) |
| Central anterior chamber depth, mm, mean ± SD (range) | 3.20±0.38 (2.23–3.84) |
| Lens status, phakic number (%) | 35 (100) |
| Shaffer's angle width grade, mean ± SD (range) | 3.29±0.84 (1–4) |
| Scheie's angle pigmentation grade, mean ± SD (range) | 1.13±0.74 (0–3) |
| Normal, number (%) | 5 (14) |
| OH, number (%) | 5 (14) |
| POAG patients, number (%) | 10 (29) |
| PEG patients, number (%) | 5 (14) |
| SOAG patients, number (%) | 5 (14) |
| PACG patients, number (%) | 5 (14) |
| Image quality grading, mean ± SD (range) | 0.33±0.47 (0–1) |

LogMAR = logarithm of the minimum angle of resolution; OH = ocular hypertension; POAG = primary open-angle glaucoma; PEG = pseudoexfoliation glaucoma; SOAG = secondary open-angle glaucoma; PACG = primary angle-closure glaucoma; SD = standard deviation.

tau coefficients were 0.47, 0.65, 0.39 and 0.62 (P<0.01, respectively). Moreover, we also analyzed the agreements for closed angle detection in each sector (n = 140) and in each eye (n = 35), and the Fleiss' kappa (95% CI) were 0.72 (0.49–0.95) and 1.00 (1.00–1.00). Additionally, the Kendall's tau coefficients were 0.75 and 1.00 (P<0.01, respectively). S1 Fig shows the radar charts of the distributions of iridocorneal angle evaluations with manual gonioscopy and with automated gonioscope for visualizing the variabilities. S1 Table demonstrates the comparison of manual gonioscopy and automated gonioscope with contingency table.

## Intraobserver agreement and correlation analyses for angle evaluations using GS-1 gonio-photos

Table 3 shows the intraobserver agreement values of the independent ophthalmologists (observers 1, 2, 3, 4, and 5) including the glaucoma specialists (observers 1, 2, and 3). One hundred forty images obtained from the four ocular sectors of 35 eyes were evaluated, and the kappa agreements between the first and second tests were evaluated. Only observer 3 was inexperienced using the Gonioscope GS-1 in the clinic. The intraobserver agreement values for the four elements, i.e., Scheie's angle width and pigmentation gradings and for detection of PAS and the Sampaolesi line, by the three glaucoma specialists ranged from 0.32 to 0.65, 0.24 to 0.71, 0.35 to 0.70 (all fair to substantial), and 0.20 to 0.76 (slight to substantial), respectively. In addition, the intraobserver agreement values for the five ophthalmologists ranged from 0.32 to 0.65 and 0.24 to 0.71 (both fair to substantial), 0.35 to 1.00 (fair to almost perfect), and -0.01 to 0.76 (poor to substantial), respectively. The glaucoma specialists did not always have higher Fleiss' kappa coefficients than the other ophthalmologists, while the intraobserver agreement

**Table 2. Agreement analyses for angle evaluations between manual gonioscopy and automated gonioscope.**

| Parameters | Manual gonioscopy (mean ± SD) | Automated gonioscope (mean ± SD) | Wilcoxon signed rank test (P) | Fleiss' kappa coefficient | | Kendall rank correlation coefficient | |
|---|---|---|---|---|---|---|---|
| | | | | Kappa (95% CI) | Landis-Koch score* | Tau | P |
| Scheie's angle width grading, n = 140 | 0.71 ± 0.89 | 0.87 ± 1.08 | 0.09 | 0.22 (0.10–0.35) | Fair agreement | 0.47 | <0.01 |
| Scheie's angle pigmentation grading, n = 140 | 1.13 ± 0.90 | 1.41 ± 1.03 | <0.01 | 0.40 (0.28–0.51) | Fair agreement | 0.65 | <0.01 |
| PAS detection, n = 140 | 0.04 ± 0.20 | 0.10 ± 0.29 | <0.01 | 0.32 (0.03–0.61) | Fair agreement | 0.39 | <0.01 |
| Sampaolesi line detection, n = 140 | 0.06 ± 0.25 | 0.12 ± 0.33 | 0.01 | 0.58 (0.35–0.81) | Moderate agreement | 0.62 | <0.01 |

PAS = peripheral anterior synechia; SD = standard deviation; CI = confidence interval.

*The Landis-Koch score is used to interpret the kappa coefficient.

values of the glaucoma specialists were the highest except for PAS detection. Additionally, the two glaucoma specialists (observers 1 and 2) who had experience using the Gonioscope GS-1 in the clinic tended to have higher scores than the other (observer 3) in all the angle evaluations. Fig 1 shows the radar charts of the distributions of iridocorneal angle evaluations from the first and second tests by the three glaucoma specialists for visualizing the intraobserver variabilities. Moreover, the Kendall's tau coefficients for angle width gradings were 0.48 to 0.81 (P<0.01, respectively) and for angle pigmentation gradings were 0.48 to 0.82 (P<0.01, respectively), which reflected the positive correlations in the evaluations (Table 4). S2 Table demonstrates the comparison of Scheie's angle gradings by glaucoma specialists with automated gonioscope between the first and second tests with contingency table.

## Interobserver agreement and correlation analyses for angle evaluations using GS-1 gonio-photos

Table 5 shows the interobserver agreement values among the independent glaucoma specialists (observers 1, 2, and 3) and among the five ophthalmologists (observers 1, 2, 3, 4, and 5) for the angle evaluations using gonioscopic photos during the first test. One hundred forty images obtained from four ocular sectors of 35 eyes were evaluated, and the kappa agreements were tested. The kappa coefficients of reliability for Scheie's angle width and pigmentation gradings and for detection of PAS and the Sampaolesi line among the three glaucoma specialists were 0.31, 0.38, and 0.31 (all fair), and 0.17 (slight), respectively. In addition, the kappa coefficients of reliability for the agreement values among the five ophthalmologists were 0.17 (slight agreement) and 0.34 (fair agreement) and 0.09, and 0.14 (both slight), respectively. Overall, the Fleiss' kappa coefficients among the three glaucoma specialists were higher than those among the five ophthalmologists. Fig 2 shows the radar charts of the distributions of the iridocorneal angle evaluations from the first test by the three glaucoma specialists for visualizing the interobserver variabilities. Moreover, we also performed the interobserver agreement and correlation analyses between the two glaucoma specialists each for the angle gradings using gonioscopic photos during the first test. The kappa coefficients for angle width and pigmentation gradings were 0.30 to 0.35 (fair agreement each), and 0.29 to 0.43 (fair to moderate), which showed the similar values. The Kendall's tau coefficients were 0.53 to 0.65 (P<0.01,

**Table 3. Intraobserver agreement analyses for angle evaluations using gonioscopic photos obtained using the GS-1.**

| Parameters | Total (n = 140) | | Superior position (n = 35) | Temporal position (n = 35) | Inferior position (n = 35) | Nasal position (n = 35) |
|---|---|---|---|---|---|---|
| | Fleiss' kappa coefficient (95% CI) | Landis-Koch score* | Fleiss' kappa coefficient (95% CI) | Fleiss' kappa coefficient (95% CI) | Fleiss' kappa coefficient (95% CI) | Fleiss' kappa coefficient (95% CI) |
| **Scheie's angle width grading** | | | | | | |
| Observer 1 | 0.65 (0.55–0.75) | Substantial agreement | 0.76 (0.57–0.92) | 0.67 (0.46–0.86) | 0.55 (0.30–0.76) | 0.60 (0.34–0.82) |
| Observer 2 | 0.63 (0.51–0.76) | Substantial agreement | 0.78 (0.44–1.00) | 0.75 (0.49–0.94) | 0.40 (0.16–0.64) | 0.63 (0.34–0.90) |
| Observer 3 | 0.32 (0.17–0.46) | Fair agreement | 0.68 (0.42–0.89) | 0.19 (0.00–0.34) | 0.15 (-0.14–0.43) | -0.03 (-0.23–0.17) |
| Observer 4 | 0.50 (0.30–0.65) | Moderate agreement | 0.25 (-0.10–0.55) | 0.50 (0.09–0.78) | 0.85 (-0.01–1.00) | 0.53 (0.11–0.84) |
| Observer 5 | 0.40 (0.26–0.53) | Fair agreement | 0.41 (0.13–0.66) | 0.37 (0.05–0.64) | 0.60 (0.34–0.81) | 0.17 (-0.09–0.42) |
| **Scheie's angle pigmentation grading** | | | | | | |
| Observer 1 | 0.67 (0.56–0.76) | Substantial agreement | 0.61 (0.36–0.81) | 0.53 (0.27–0.73) | 0.53 (0.30–0.74) | 0.87 (0.69–1.00) |
| Observer 2 | 0.71 (0.59–0.82) | Substantial agreement | 0.80 (0.46–1.00) | 0.58 (0.18–0.87) | 0.70 (0.48–0.88) | 0.67 (0.31–0.90) |
| Observer 3 | 0.24 (0.13–0.36) | Fair agreement | 0.28 (0.00–0.52) | 0.43 (0.15–0.64) | 0.05 (-0.19–0.27) | 0.07 (-0.19–0.31) |
| Observer 4 | 0.65 (0.54–0.75) | Substantial agreement | 0.65 (0.41–0.84) | 0.79 (0.55–0.95) | 0.57 (0.31–0.79) | 0.47 (0.18–0.69) |
| Observer 5 | 0.45 (0.30–0.57) | Moderate agreement | 0.47 (0.16–0.73) | 0.49 (0.16–0.75) | 0.37 (0.10–0.63) | 0.40 (0.10–0.65) |
| **PAS detection** | | | | | | |
| Observer 1 | 0.60 (0.32–0.82) | Moderate agreement | 0.77 (0.30–1.00) | 0.63 (-0.06–1.00) | 0.20 (-0.15–0.67) | 1.00 (1.00–1.00) |
| Observer 2 | 0.70 (0.32–1.00) | Substantial agreement | 0.65 (-0.05–1.00) | 1.00 (1.00–1.00) | 0.72 (-0.02–1.00) | -0.02 (-0.06–0.01) |
| Observer 3 | 0.35 (0.08–0.58) | Fair agreement | 0.44 (-0.09–0.87) | 0.64 (-0.04–1.00) | 0.11 (-0.23–0.46) | -0.01 (-0.06–0.01) |
| Observer 4 | 1.00 (1.00–1.00) | Almost perfect agreement | 1.00 (1.00–1.00) | NA | NA | NA |
| Observer 5 | 0.51 (0.34–0.67) | Moderate agreement | 0.62 (-0.04–1.00) | 0.60 (0.15–0.92) | 0.77 (0.44–1.00) | 0.12 (-0.26–0.42) |
| **Sampaolesi line detection** | | | | | | |
| Observer 1 | 0.60 (0.39–0.77) | Moderate agreement | NA | -0.03 (-0.08–0.01) | 0.22 (-0.12–0.49) | NA |
| Observer 2 | 0.76 (0.39–1.00) | Substantial agreement | NA | NA | 0.72 (0.30–1.00) | NA |
| Observer 3 | 0.20 (-0.02–0.43) | Slight agreement | NA | 0.07 (-0.21–0.46) | 0.21 (-0.16–0.51) | -0.08 (-0.17–-0.03) |
| Observer 4 | -0.01 (-0.03–0.00) | Poor agreement | NA | -0.03 (-0.08–0.01) | -0.03 (-0.08–0.01) | NA |
| Observer 5 | 0.53 (-0.01–0.89) | Moderate agreement | NA | 0.64 (-0.04–1.00) | 0.35 (-0.09–1.00) | NA |

Observer 1, 2, 3 = glaucoma specialists.

Observer 4, 5 = ophthalmologists.

PAS = peripheral anterior synechia; CI = confidence interval; NA = not applicable.

*The Landis-Koch score is used to interpret the kappa coefficient.

respectively) and 0.62 to 0.69 (P<0.01, respectively), which reflected the positive correlations between the two each (Table 6). S3 Table demonstrates the comparison of Scheie's angle gradings with automated gonioscope between a glaucoma specialist and the others in first test with contingency table. We additionally assessed the effects of GS-1 image quality on the observer

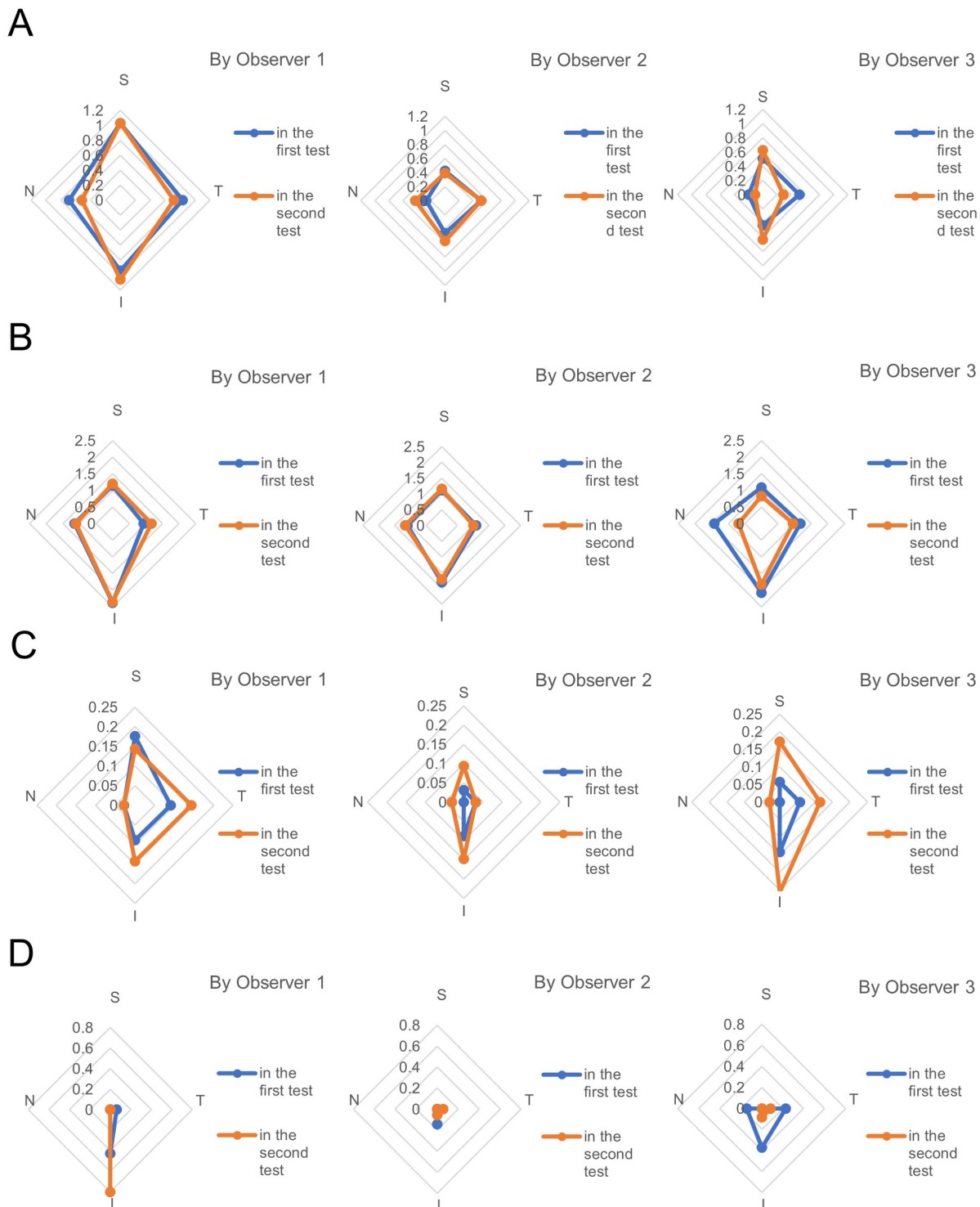

**Fig 1. The radar charts of the mean distributions of iridocorneal angle evaluations during the first and second tests by three independent glaucoma specialists for visualizing intraobserver variabilities in (A) Scheie's angle width grading (0 to 4), (B) Scheie's angle pigmentation grading (0 to 4), (C) PAS detection (0 to 1), and (D) Sampaolesi line detection (0 to 1).** S = superior; T = temporal; I = inferior; N = nasal.

**Table 4. Intra-observer correlation analyses for angle gradings using gonioscopic photos of GS-1.**

| Parameters | Kendall rank correlation coefficient (total, n = 140) | | Kendall coefficient (superior position, n = 35) | | Kendall coefficient (temporal position, n = 35) | | Kendall coefficient (inferior position, n = 35) | | Kendall coefficient (nasal position, n = 35) | |
|---|---|---|---|---|---|---|---|---|---|---|
| | Tau | P | Tau | P | Tau | P | Tau | P | Tau | P |
| **Scheie's angle width grading** | | | | | | | | | | |
| Observer 1 | 0.81 | <0.01 | 0.88 | <0.01 | 0.87 | <0.01 | 0.75 | <0.01 | 0.78 | <0.01 |
| Observer 2 | 0.75 | <0.01 | 0.81 | <0.01 | 0.82 | <0.01 | 0.66 | <0.01 | 0.75 | <0.01 |
| Observer 3 | 0.48 | <0.01 | 0.84 | <0.01 | 0.61 | <0.01 | 0.21 | 0.17 | 0.12 | 0.50 |
| Observer 4 | 0.60 | <0.01 | 0.52 | <0.01 | 0.68 | <0.01 | 0.86 | <0.01 | 0.53 | <0.01 |
| Observer 5 | 0.52 | <0.01 | 0.64 | <0.01 | 0.51 | <0.01 | 0.65 | <0.01 | 0.27 | 0.09 |
| **Scheie's angle pigmentation grading** | | | | | | | | | | |
| Observer 1 | 0.82 | <0.01 | 0.77 | <0.01 | 0.68 | <0.01 | 0.74 | <0.01 | 0.93 | <0.01 |
| Observer 2 | 0.81 | <0.01 | 0.83 | <0.01 | 0.72 | <0.01 | 0.84 | <0.01 | 0.75 | <0.01 |
| Observer 3 | 0.48 | <0.01 | 0.51 | <0.01 | 0.57 | <0.01 | 0.07 | 0.64 | 0.44 | <0.01 |
| Observer 4 | 0.79 | <0.01 | 0.79 | <0.01 | 0.82 | <0.01 | 0.75 | <0.01 | 0.74 | <0.01 |
| Observer 5 | 0.54 | <0.01 | 0.63 | <0.01 | 0.59 | <0.01 | 0.47 | <0.01 | 0.55 | <0.01 |

Observer 1, 2, 3 = glaucoma specialists.

Observer 4, 5 = ophthalmologists.

**Table 5. Interobserver agreement analyses for angle evaluations using GS-1 gonioscopic photos during the first test.**

| Parameters | Total (n = 140) | | Superior position (n = 35) | Temporal position (n = 35) | Inferior position (n = 35) | Nasal position (n = 35) |
|---|---|---|---|---|---|---|
| | Fleiss' kappa coefficient (95% CI) | Landis-Koch score* | Fleiss' kappa coefficient (95% CI) | Fleiss' kappa coefficient (95% CI) | Fleiss' kappa coefficient (95% CI) | Fleiss' kappa coefficient (95% CI) |
| **Scheie's angle width grading** | | | | | | |
| Among three glaucoma specialists | 0.31 (0.22–0.40) | Fair agreement | 0.32 (0.14–0.48) | 0.43 (0.31–0.52) | 0.36 (0.17–0.53) | 0.07 (-0.09–0.23) |
| Among all five ophthalmologlists | 0.17 (0.11–0.23) | Slight agreement | 0.16 (0.03–0.28) | 0.17 (0.05–0.24) | 0.19 (0.05–0.31) | 0.12 (0.02–0.21) |
| **Scheie's angle pigmentation grading** | | | | | | |
| Among three glaucoma specialists | 0.38 (0.30–0.46) | Fair agreement | 0.26 (0.05–0.46) | 0.44 (0.20–0.62) | 0.23 (0.07–0.38) | 0.44 (0.25–0.60) |
| Among all five ophthalmologlists | 0.34 (0.28–0.40) | Fair agreement | 0.34 (0.17–0.48) | 0.29 (0.11–0.43) | 0.24 (0.13–0.33) | 0.40 (0.24–0.51) |
| **PAS detection** | | | | | | |
| Among three glaucoma specialists | 0.31 (0.09–0.49) | Fair agreement | 0.11 (-0.08–0.38) | 0.65 (-0.08–0.38) | 0.22 (-0.04–0.39) | -0.01 (-0.04–0.01) |
| Among all five ophthalmologlists | 0.09 (0.01–0.17) | Slight agreement | 0.12 (0.03–0.19) | 0.19 (-0.05–0.37) | 0.22 (0.05–0.35) | -0.13 (-0.17–0.09) |
| **Sampaolesi line detection** | | | | | | |
| Among three glaucoma specialists | 0.17 (-0.01–0.35) | Slight agreement | NA | -0.11 (-0.19–0.05) | 0.12 (-0.15–0.34) | -0.05 (-0.10–0.01) |
| Among all five ophthalmologlists | 0.14 (0.01–0.26) | Slight agreement | NA | 0.04 (-0.05–0.13) | 0.09 (-0.09–0.23) | -0.03 (-0.06–-0.01) |

PAS = peripheral anterior synechia; CI = confidence interval; NA = not applicable.

*The Landis-Koch score is used to interpret the kappa coefficient.

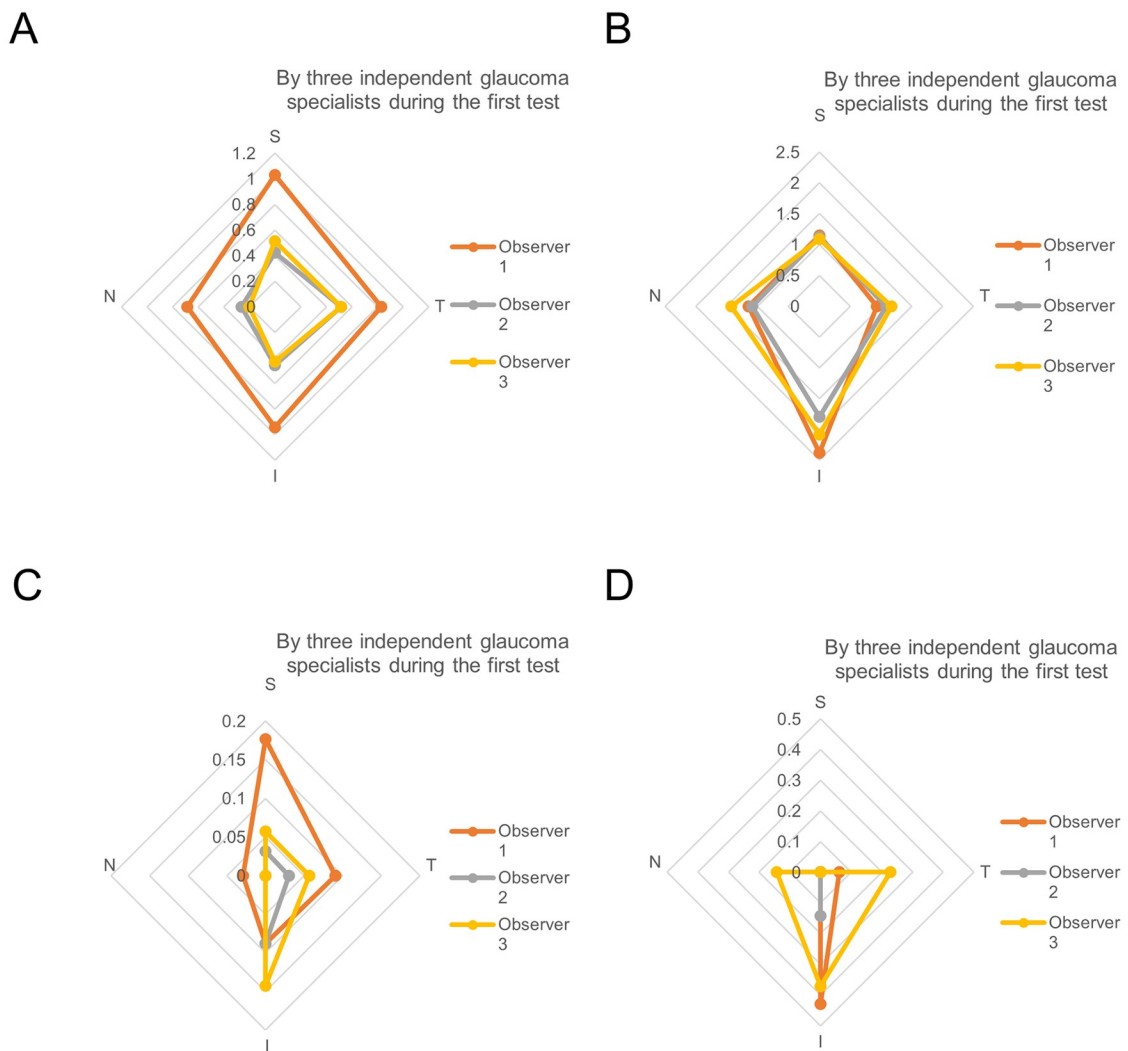

**Fig 2. The radar charts of the mean distributions of iridocorneal angle evaluations during the first test by three independent glaucoma specialists for visualizing interobserver variabilities in (A) Scheie's angle width grading (0 to 4), (B) Scheie's angle pigmentation grading (0 to 4), (C) PAS detection (0 to 1), and (d) Sampaolesi line detection (0 to 1).** S = superior; T = temporal; I = inferior; N = nasal.

agreements for angle evaluations. As the result, the observer agreements using grade 0 images seems not always better than those using grade 1 images (S4 Table).

## Discussion

Using the images obtained by the Gonioscope GS-1, we evaluated the intraobserver and inter-observer agreement values of five independent ophthalmologists including three glaucoma specialists in multiple centers for several iridocorneal angle evaluations in four ocular sectors. Fleiss' kappa statistic was used to demonstrate the observer agreements for angle evaluations with GS-1 images because the exact match is desirable in clinical evaluations. In addition, we also performed the Kendall rank correlation coefficient analyses especially for the ordinal scales, which would be beneficial to assess the statistical associations based on the ranks of the data.

**Table 6. Inter-observer reproducibility analyses for angle gradings by glaucoma specialists using gonioscopic photos of GS-1.**

| Parameters | Fleiss' kappa coefficient (total, n = 140) | | Kendall rank correlation coefficient (total, n = 140) | |
|---|---|---|---|---|
| | Kappa (95% CI) | Landis-Koch score* | Tau | P |
| **Scheie's angle width grading** | | | | |
| Observer 1 vs. Observer 2 | 0.31 (0.20 to 0.42) | Fair agreement | 0.65 | <0.01 |
| Observer 1 vs. Observer 3 | 0.30 (0.19 to 0.41) | Fair agreement | 0.57 | <0.01 |
| Observer 2 vs. Observer 3 | 0.35 (0.21 to 0.49) | Fair agreement | 0.53 | <0.01 |
| **Scheie's angle pigmentation grading** | | | | |
| Observer 1 vs. Observer 2 | 0.40 (0.29 to 0.52) | Fair agreement | 0.69 | <0.01 |
| Observer 1 vs. Observer 3 | 0.43 (0.31 to 0.54) | Moderate agreement | 0.65 | <0.01 |
| Observer 2 vs. Observer 3 | 0.29 (0.17 to 0.41) | Fair agreement | 0.62 | <0.01 |

Observer 1, 2, 3 = glaucoma specialists.

Manual gonioscopy, a contact method developed in the 1800s that requires topical anesthesia and patient cooperation, is the current clinical standard for angle assessment because we can perform both static and dynamic angle evaluations with it, which enables us to distinguish between an anatomically closed angle with iridotrabecular contact (ITC, apposition) and PAS. However, gonioscopy is subjective, and findings may vary with corneal pressure, lighting conditions, angle pigmentation, and iris convexity. Therefore, the interobserver (intervisit) agreement for angle opening with binary scale (narrow or open) with gonioscopy was moderate to excellent, 0.53 to 0.86 and the interobserver agreement was moderate to substantial, 0.57 to 0.69, which was calculated with kappa statistics [24]. In addition, the interobserver agreement for angle width gradings between the resident's observations and the glaucoma specialist by manual gonioscopy was substantial, 0.75, which was determined with weighted kappa statistics [25]. The lack of objective documentation of the gonioscopic findings makes it inappropriate for long-term follow-up. Considering this, the angle evaluations using recordable standardized gonio-photos may contribute to more reliable and reproducible assessments. However, the reproducibility of the angle evaluations using manual standardized gonio-photos has not always been high. Phu et al. reported that the agreement values for angle width evaluations between the results of gonioscopy performed by a senior optometrist and gonioscopic image evaluations by three experienced optometrists were 0.39 to 0.45 (fair to moderate agreement); the interobserver agreement values for the evaluations with gonio-photography between the two were 0.42 to 0.76 (moderate to substantial agreement), which were analyzed by weighted Fleiss' kappa [18]. Therefore, manual evaluations and gonioscopic image evaluations with conventional gonioscopy have limitations, respectively.

While, the inter- and intraobserver agreement for EyeCam which is the other angle visualization and photography device was excellent in angle closure detection (kappa = 0.82 and 0.87), and the intraobserver agreement between gonioscopy and EyeCam in angle closure detection was moderate (kappa = 0.52–0.60), which were calculated with unweighted kappa. The intraobserver agreement between gonioscopy and EyeCam in angle width grading was moderate (kappa = 0.52–0.60, calculated with weighted kappa) [16]. However, participants must be performed EyeCam imaging in suprine position with the 130°direct gonio-lens, which takes longer than gonioscopy, approximately 5 to 10 minutes per eye and the image is quite different from the common manual indirect gonioscopy, which might be the reasons that the device is not used in common. Moreover, it is also disadvantageous in terms of acquiring standard images. Unlike the GS-1's automated angle imaging system, there are many

manual operations such as adjusting the illumination condition, position and tilt of gonio-scopy lens for photographing with EyeCam, which could affect the angle evaluations [13, 16].

As shown in Table 2, compared with the results of manual gonioscopy, the outcomes using GS-1 gonio-photos tended to be overestimated. In addition, the Fleiss' kappa values for angle width grading, and for detection of PAS and the Sampaolesi line by the glaucoma specialist were different from the previous results of a prototype version of GS-1 (kappa = 0.48, 0.46 and 0.09, respectively) [9]. Additionally, the Kendall's tau coefficients represented the positive correlations between the two methods. Gonioscopy allows us to perform not only conventional static but also dynamic gonioscopy, and we can observe the part we want to see in more detail until we are able to judge. Therefore, the assessment with gonioscopy by the glaucoma specialist would be different from that with the single static GS-1 image and could have better diagnostic power than it. On the other hand, the Fleiss' kappa values for closed angle detection with GS-1 in each sector and in each eye were better than the prototype GS-1 and might be better than EyeCam [9, 16]. The high agreement for closed angle detection supported the usefulness of the device which could be suitable for the screening because, for the purpose of screening, as in fundus photography that is commonly used in medical checkup, the overestimation would be less of a problem. The important thing is to operate the two methods for appropriate purposes.

Table 3 would reflect the certain views on the intraobserver agreements of the independent ophthalmologists including the glaucoma specialists and the existence of a large measurement bias in the method. In addition, we could find the similar tendency in each sector analysis, however it would be difficult to conclude the findings also holds in each sector because of the large differences of value and confidence interval. While, the Kendall's tau coefficients represented the positive correlations between the outcomes of first and second test in total and in most of each sector analysis (Table 4). To the best of our knowledge, no study has evaluated the intraobserver agreement in angle evaluations using standardized gonio-photos with indirect gonioscopy; however, even among the glaucoma specialists in the current study, the intraobserver agreement values were not always substantial or higher scores. The manual gonioscopy technique may have a longer-than-desired learning curve [9]. Therefore, as with conventional gonioscopy, angle evaluations using the gonio-photos also likely have an associated learning curve that can affect the results. In fact, the two glaucoma specialists who were experienced using the Gonioscope GS-1 tended to have higher scores than the other. This suggested that even highly experienced glaucoma specialists need additional training to assess the angles using the gonio-images in clinical settings; this can be accomplished by comparing the actual slit-lamp findings and manual gonioscopy findings before evaluating the angles only using the gonio-photos, which could be a hidden potential limitation of the device. Moreover, the two ophthalmologists who were not glaucoma specialists experienced in the use of the Gonioscope GS-1 tended to have higher intraobserver agreement scores in the angle gradings than the inexperienced glaucoma specialist, likely also for the same reason. In addition, if using the gonio-photos, the angle assessments, especially in the presence of ambiguous findings, may be difficult even for the experts. Moreover, when considering PAS detection by observer 4, for example, in the presence or absence of specific findings, a tendency may exist for overestimation or underestimation depending on the observer. Thus, careful interpretation is important since the apparent match rate is high in the cases.

Overall, the Fleiss' kappa coefficients among the three glaucoma specialists were higher than those among the five ophthalmologists and the interobserver agreements were relatively lower than those of the intraobserver agreements (Table 5), which could have been affected by some differences in angle recognition among the different ophthalmologists. In addition, the interobserver agreements improved according to the proportion of the glaucoma specialists

among the observers. While, the Kendall's tau coefficients between the two glaucoma specialists each represented the positive correlations (Table 6). Therefore, as expected, the angle assessments by glaucoma specialists were the most similar and thus the most reliable. We could find the similar tendency in each sector analysis; however, it would be difficult to conclude the findings also holds in each sector because of the large differences of value and confidence interval. Additionally, even the current agreement among the three glaucoma specialists and between the two each did not always achieve high scores. The interobserver agreement values for Scheie's angle pigmentation grading were similar between the observer groups. Therefore, the grading is easier, and relatively good accuracy can be achieved from the beginning. In addition, the agreement for detecting Sampaolesi lines between manual gonioscopy and the gonio-image assessment was slight (kappa = 0.16), and in a previous report, the interobserver agreement and kappa coefficient between an ophthalmology resident and glaucoma specialist was slight (kappa = 0.09) [9]. Considering that the intraobserver agreement for detecting Sampaolesi lines by the glaucoma specialists tended to be high, the low interobserver agreement scores probably resulted from some differences in the recognition by the observers rather than the characteristics of the assessments with the gonio-images. Therefore, to reduce the interobserver variability in the angle evaluations with the gonio-photos, the consensus on the angle findings between the observers should be reconfirmed.

We additionally assessed the effects of GS-1 image quality on the observer agreements for angle evaluations. As shown in S4 Table, the observer agreements using grade 0 images seems not always better than those using grade 1 images, probably because we completely excluded the grade 2 (blurred with no discernible details) images. However, our study had several limitations. Because our study subjects were from university hospitals that provide special glaucoma cares on a daily basis in Japan, there may have been referral bias. Moreover, our study subjects were all Japanese. The irises and angles of Asians differ in color from those of Caucasians, which may differ from other races. We also excluded subjects with poor-quality images, which could cause selection bias. The sample size was not sufficiently large to include all angle types, and there likely is a limit to generalizability of the current results to all angle evaluations with the gonio-photos. However, it is impractical to evaluate the diagnostic reproducibility of all rare angle findings with small sample size. Therefore, we limited the evaluation items in our study. Finally, the ophthalmologists participating in the study evaluated the angles using only the gonio-photos and had no access to other information, which differs markedly from normal clinical situations and may cause misclassification bias.

In conclusion, the current study confirmed the results of previous reports and demonstrated new perspectives about the reproducibility of iridocorneal angle assessments using the gonio-images by the five independent ophthalmologists including three glaucoma specialists. As result, the high agreement between gonioscopy and Gonioscope GS-1 for closed angle detection supported the usefulness of the device for screening, however the further study must be needed. The intraobserver agreement levels for Scheie's angle width and pigmentation gradings and for detecting PAS and Sampaolesi lines among three glaucoma specialists were fair to substantial for the first three and slight to substantial for the last, while the interobserver agreements were fair for the first three and slight for the last. Our findings suggested slight-to-substantial intraobserver agreement and slight-to-fair (among the three) or fair-to-moderate (between the two each) interobserver agreement for the angle assessments using the gonio-photos even by glaucoma specialists. Generally, the intraobserver agreement levels among the glaucoma specialists tended to be high, and the interobserver agreements improved based on the proportion of the glaucoma specialists among the observers. Therefore, the angle assessments by the glaucoma specialists were the most similar and thus the most reliable. However, as with conventional gonioscopy, the angle evaluation using the gonio-photos likely is

associated with a specific learning curve and the need for additional training in clinical practice before assessments can be performed using only the gonio-photos even for glaucoma specialists. It also is necessary to reconfirm the consensus regarding the angle findings among the observers to reduce the interobserver variabilities in angle evaluations when using the gonio-photos. Sufficient training and a solid consensus should allow us to perform more reliable angle assessments using GS-1 gonio-photos with high reproducibility.

## Supporting information

**S1 Fig. The radar charts of the distributions of iridocorneal angle evaluations with manual gonioscopy and automated gonioscope by the glaucoma specialist (MT) for visualizing the variabilities in (A) Scheie's angle width grading, (B) Scheie's angle pigmentation grading, (C) PAS detection, and (D) Sampaolesi line detection.**
(TIF)

**S1 Table. Comparison of manual gonioscopy and automated gonioscope in all angle gradings.**
(DOCX)

**S2 Table. Comparison of Scheie's angle gradings by glaucoma specialists with automated gonioscope between first and second tests in all images.**
(DOCX)

**S3 Table. Comparison of Scheie's angle gradings with automated gonioscope between a glaucoma specialist and the others in first test.**
(DOCX)

**S4 Table. Effect of image quality on observer agreement for angle evaluations using gonioscopic photos of GS-1.**
(DOCX)

**S1 Data. The first test of randomized 140 gonio-images.**
(PDF)

**S2 Data. The second test of different randomized 140 gonio-images.**
(PDF)

## Author Contributions

**Conceptualization:** Masato Matsuo, Masaki Tanito.

**Data curation:** Masato Matsuo, Shiro Mizoue, Koji Nitta, Yasuyuki Takai, Kazunobu Sugihara, Masaki Tanito.

**Formal analysis:** Masato Matsuo.

**Investigation:** Masato Matsuo, Masaki Tanito.

**Methodology:** Masato Matsuo, Shiro Mizoue, Koji Nitta, Masaki Tanito.

**Writing – original draft:** Masato Matsuo.

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
