## [Decision Letter · Decision Letter 0]

16 Dec 2020

PONE-D-20-24626

Intraobserver and interobserver agreement among anterior chamber angle evaluations using automated 360-degree gonio-photos

PLOS ONE

Dear Dr. Matsuo,

We look forward to receiving your revised manuscript.

Kind regards,

Jinhai Huang, M.D.

Academic Editor

PLOS ONE

Journal Requirements:

Reviewers' comments:

Reviewer's Responses to Questions

**Comments to the Author**

1. Is the manuscript technically sound, and do the data support the conclusions?

Reviewer #1: Partly

2. Has the statistical analysis been performed appropriately and rigorously? 

Reviewer #1: No

3. Have the authors made all data underlying the findings in their manuscript fully available?

Reviewer #1: No

4. Is the manuscript presented in an intelligible fashion and written in standard English?

Reviewer #1: Yes

5. Review Comments to the Author

Reviewer #1: Matsuo and colleagues reported on the agreement of anterior chamber angle evaluations using the GS-1. This is an emerging technique and thus this kind of study is useful in contributing to the literature. There are a number of primary concerns regarding the writing and the analysis approach that the authors could consider, as well as aspects of the discussion that appear to be lacking in the requisite depth for a critical review of the technology.

Introduction:

1) The first two paragraphs appear to be needlessly long: the authors could consider reshaping this to identify two succinct points: 1) angle closure as a disease entity is important to identify as it has a different prognostic course compared to open angle glaucoma; 2) secondary open angle (and closed anglee) glaucomas require examination of the anterior chamber angle to identify underlying causes that may alter the treatment plan. Paragraph two in particular is a bit confusing in its tone with regard to the usefulness and limitations of gonioscopy. Moreover, there are some aspects of gonioscopy that are glossed over, such as the fact that grading systems for treatment titration rely solely on the gonioscopic impression (e.g. Prum Jr et al 2016 Ophthalmology), and that major clinical trials use gonioscopy as the technique of choice (He et al 2019 Lancet). Gonioscopy lenses such as the G6 in theory offer a more uninterrupted view of the anterior chamber angle so the statement on page 6, line 10 is not quite true.

2) Aside from the pigmentation grade, the configuration and distribution of pigment is important clinically, such as in cases of burnt out pigment dispersion syndrome, mottled versus homogenous pigmentation - there are numerous citations for this, such as from Rob Ritch.

Methods:

1) Why did the authors use Scheie classification in the first instance (page 8), followed by Shaffer (page 9)? Both are different and provide the opposite ordinal grades. This was unclear, but suggests that one was used by the clinician for the assessment and the other for the experimental design? Needs clarification as to the purpose of the grading system.

2) Page 10: the reader is slightly confused with the exclusion process... was the whole eye excluded or was it just one image per eye that was excluded - based on lines 14-16. How was the better quality image determined if both were equal? Were some angles systematically more likely to have better images? For example, if considering conventional imaging modalities, it is not surprising to find that superior angle assessment is more problematic compared to other quadrants due to anatomic lid interference (e.g. Xu et al 2019 TVST).

3) A limitation of the GS-1, like AS-OCT is that it is performed in primary gaze without lens tilt, despite many clinical trials adding the dynamic component of gonioscopy (e.g. He et al 2019 Lancet) and clinical guidelines recommending identification of iridotrabecular contact that is only possible with lens tilt or off-axis gaze (e.g. Prum Jr et al 2016 Ophthalmology). This requires a comment on page 11.

4) The methods require a bit more transparency in the writing. Pages 12-13 for example appear particularly disorganised. The reader is introduced to five independent observers firstly on page 12, but then on page 13 line 3 onwards, a presumably singular observer is used for the intraobserver evaluation. However, then the authors state that a "second set" was used to further evaluate intraobserver agreements presumably for the five independent observers on line 16? Another example is page 13 line 13 - are the different randomised images from the same set or another prospectively collected set? The writing needs further clarification and would perhaps benefit from separation into subheadings that allow clarity in the methods.

Results

1) There are a few issues regarding the presentation of Table 1. The authors did not seem to state the techniques used to measure many of the continuous variables reported in the table. Further to Table 1, why was pseudoexfoliative glaucoma and secondary open angle glaucoma separated? The reporting of the distribution of Shaffer's angle width but Scheie's angle pigmentation grade represents to the reader unnecessary differentiation and muddling of the grading schemes - why not just stick to one? Further, it could be argued that the reading of a mean and SD for these grades is questionable, due to their non-linearity and ordinal nature (see Phu et al 2020 OPO)... it may be more useful to break down the distribution of the exact numbers of each grade, especially since the Fleiss's kappa was used. Finally, the additive value of topical glaucoma medications is questionable: the paper is not really reporting on the contribution of these to angle grading and the hypothesis being tested is unlikely to be confounded by these factors. It would also be more informative to provide the proportion of poor image grades that were subsequently excluded from analysis in this table or in the results text. This would reinforce the authors' choice of a pragmatic approach and a more realistic impression of the deployment of GS-1.

2) With regard to reporting Table 2 and the kappa values, it is unclear to the reader why Fleiss's kappa was used for binary variables (PAS and Sampaolesi line - present/absent). This confusion stems from the lack of contingency tables being presented in the results and the confusing methods. The authors should seek to clarify the most appropriate statistical method used. Fleiss's kappa may be appropriate for agreement between the observers but from the writing it is hard to say what was compared.

3) The reporting of closed angle detection is somewhat unclear and strikes the reader as somewhat misleading and requires clarification in writing. Not all of the 140 quadrants were closed, and so the question is whether the kappa was related to the binary outcome of "open" vs. "closed", or the agreement between observers (as noted above).

4) Page 17, line 14 refers to observers 1-5 and glaucoma specialists 1-3, but this was somewhat unclear from the methods as well.

5) Table 3 (and similar): there was extreme variability in the kappa values across quadrants and across observers: could the authors comment on whether there was a systematic difference or random difference in the text? This is a very interesting result and the reader may be tempted to think it may be related to the distribution of angle grades, whereby some may be more obvious and easy to score (e.g. closed or very wide open angles) compared to the middling grades.

6) Overall, comments 3-5 above suggest that contingency tables may play a role in the reporting of the data

Discussion

1) The second paragraph of the discussion is confusing, and the reader is unsure what the authors are trying to say. The tone is confusing in whether 10% is a high number or a low number?

2) With regard to objectivity, one questions the value of precise grading given the wealth of data required to change the management plan. For example, the decision to treat would be based on a multitude of factors aside from the angle appearance, including accessibility to health care, age, lens status, epidemiological risk factors and others (see Thomas and Walland 2013 CEO).

3) Page 27, lines 6-11: with respect to the discussion on gonio photographs, there is a limitation previously noted in the literature that the authors seemed to have omitted: the fact that many of these are conducted in primary gaze, and lack the dynamism of manual gonioscopy, which is really the key advantage of the technique. This requires further discussion.

4) A general question with regards to the comparison with the EyeCam, the light source for imaging and photography should be mentioned or discussed at some point. There are significant efforts taken in standardising gonioscopy as a procedure to mitigate introduction of stray light through the pupil.

5) Another general comment with the discussion: there is far too much re-reporting of the results and this causes bloat in the discussion. The authors should revise this for succinctness.

6) The issue of angle grade distribution was mentioned on page 31, but the analysis seems to be lacking in terms of accounting for it as a confounding factor.

7) Another limitation of the grading system for image quality, which should also be noted in the references cited by the authors, is that the entirety of the image needs not be clear for evaluation. This does not seem to be captured with this method and should be discussed.

8) The authors raise an interesting point about the learning curve and the value of experience. This was discussed by Phu and colleagues and may be worthwhile integrated as part of this discussion (Phu et al 2019 OVS). What they may consider also acknowledging is the acquisition step. Though the processes described in the methods appear to be largely automated, there may be some component of interpreting whether the image is useable. A pragmatic approach as described by the authors would necessitate consideration of practical aspects such as the limitations and barriers to successful image acquisition and whether there are systematic errors in that process. A consecutive sampling strategy would be necessary as well as reporting of the errors associated with the technique. Finally, a calibration exercise would be suitable for this particular purpose.

Minor comments

- Page 12, line 9-10: please clarify that the participants are the ophthalmologists

- Page 13, line 22: nominal or ordinal?

- Table 1: Dioptor should be "Diopter"

- There is some inconsistency between "sectors" and "quadrants" used throughout the manuscript

- Page 25: there is some inconsistency in the way that the results are reported in the text. It is quite wordy and somewhat repetitive from the tables. The authors should note this trend throughout the manuscript and seek to clarify and write succinctly.

- Page 25, line 15: there should be a clearer distinction between an ophthalmologist (general?) and glaucoma specialist

- Why were there "NA" results in Supp Table 1 - was not clear

- The units on Figures 1 and 2 are not clearly described in the Figure Captions. Although somewhat intuitive from the text, this may deserve further clarification.

References:

Ritch Am J Ophthalmol. 1998 Sep;126(3):442-5.

Phu et al Ophthalmic Physiol Opt. 2020 Sep;40(5):617-631.

Xu et al Transl Vis Sci Technol . 2019 Mar 26;8(2):52019

Prum Jr et al Ophthalmology. 2016 Jan;123(1):P1-P40.

Thomas and Walland Clin Exp Ophthalmol. 2013 Apr;41(3):282-92.

He et al Lancet. 2019 Apr 20;393(10181):1609-1618.

Phu et al Optom Vis Sci. 2019 Oct;96(10):751-760.

---

## [Author Response · Author response to Decision Letter 0]

7 Jan 2021

Ref: PONE-D-20-24626

Intraobserver and interobserver agreement among anterior chamber angle evaluations using automated 360-degree gonio-photos

PLOS ONE

Dear Dr. Huang,

We appreciate your reconsideration of our manuscript entitled “Intraobserver and interobserver agreement among anterior chamber angle evaluations using automated 360-degree gonio-photos” for publication in PLoS One as a Full Paper.

We disagree with some of the reviewer's comments, however we received constructive feedback. We have revised our manuscript and made a point-by-point response to the comments as follows. We have worked hard to incorporate your feedback and hope that these revisions persuade you to accept our submission.

We state that this manuscript has not been published elsewhere and is not under consideration by another journal. All authors have approved the manuscript and agree with submission to PLOS ONE. Thank you!

Best regards,

Masato Matsuo, MD, PhD

Department of Ophthalmology,

Shimane University Faculty of Medicine,

Enya 89-1, Izumo, Shimane, JAPAN.

mmpeaceful@yahoo.ne.jp

matsuondmc@gmail.com

Journal Requirements:

Response

We have revised our manuscript to meet the PLOS ONE's style requirements.

Response

We have deleted our ethics statement from our title page.

Reviewer #1’s Comments:

Introduction:

1) The first two paragraphs appear to be needlessly long: the authors could consider reshaping this to identify two succinct points: 1) angle closure as a disease entity is important to identify as it has a different prognostic course compared to open angle glaucoma; 2) secondary open angle (and closed angle) glaucomas require examination of the anterior chamber angle to identify underlying causes that may alter the treatment plan. Paragraph two in particular is a bit confusing in its tone with regard to the usefulness and limitations of gonioscopy. Moreover, there are some aspects of gonioscopy that are glossed over, such as the fact that grading systems for treatment titration rely solely on the gonioscopic impression (e.g. Prum Jr et al 2016 Ophthalmology), and that major clinical trials use gonioscopy as the technique of choice (He et al 2019 Lancet). Gonioscopy lenses such as the G6 in theory offer a more uninterrupted view of the anterior chamber angle so the statement on page 6, line 10 is not quite true.

Response 

Thank you for the comment, however we disagree with you. According to the journal’s submission guideline, we tried to provide the background that puts the manuscript into context and allows readers outside the field to understand the purpose and significance of the study. The readers are not limited to glaucoma specialists or ophthalmologists. Therefore, it is necessary to define the problem addressed and explain why it is important.

Regarding 1), we also think early detection of a narrow angle and peripheral anterior synechia (PAS) is vital to prevent primary angle-closure glaucoma (PACG) as described in paragraph 1. However, we cannot agree with your idea that angle closure is important to identify as it has a different prognostic course compared to open angle glaucoma, because we sometimes experience cases of open-angle glaucoma (OAG) with end-stage visual field impairment at the first visit. If we could detect such patients early enough, we surely prevent the visual field loss due to OAG. Therefore, the early OAG detection and treatment is also clinically important.

As for 2), we agree with your comment that secondary glaucomas require examination of the anterior chamber angle to identify underlying causes that may alter the treatment plan. Thus, we have added the statement in the manuscript L18-20 (pg 4). We believe that the change helps our manuscript more contextual. On the other hand, regarding the other indications, we do not agree with you. We pointed out the limitation of subjective assessments with gonioscopy in the manuscript L12-13 (pg 5). Additionally, we also know the major clinical trials use gonioscopy as the technique of choice, however the fact has nothing to do with the reliability of the test. Moreover, “gonioscopy can examine at one time only a limited contiguous portion of the iridocorneal angle” means that we can observe only a limited position of the angle with conventional gonioscopy at one time even if using G6 because of the manual nature. On the other hand, gonioscopic camera can take whole angle images simultaneously. Therefore, we pointed out the characteristics and limitations of the gonioscopy in the manuscript.

2) Aside from the pigmentation grade, the configuration and distribution of pigment is important clinically, such as in cases of burnt out pigment dispersion syndrome, mottled versus homogenous pigmentation - there are numerous citations for this, such as from Rob Ritch.

Response 

We also believe the importance of configuration and distribution of pigment; however, it is impractical to evaluate the diagnostic reproducibility of all angle findings. Pigment dispersion syndrome is clinically rare, and the evaluation is out of our main purpose. Moreover, it is difficult to evaluate the observer agreement of rare finding with small sample size practically. Therefore, we limited the evaluation items in our study.

Methods:

1) Why did the authors use Scheie classification in the first instance (page 8), followed by Shaffer (page 9)? Both are different and provide the opposite ordinal grades. This was unclear, but suggests that one was used by the clinician for the assessment and the other for the experimental design? Needs clarification as to the purpose of the grading system.

Response 

Thank you for the comment. As you pointed out, the Scheie’s angle width grading system was used for the experimental design and the Shaffer’s grading system was for the clinical assessment as we described in the manuscript L16-23 (pg 11). Moreover, following your suggestion, we have added the supplementary explanations for easy understanding in the manuscript L18 (pg 7), L14-15 (pg 8).

2) Page 10: the reader is slightly confused with the exclusion process... was the whole eye excluded or was it just one image per eye that was excluded - based on lines 14-16. How was the better quality image determined if both were equal? Were some angles systematically more likely to have better images? For example, if considering conventional imaging modalities, it is not surprising to find that superior angle assessment is more problematic compared to other quadrants due to anatomic lid interference (e.g. Xu et al 2019 TVST).

Response 

Thank you for the comment. We paraphrased the word for clarity in the manuscript L17 (pg 9).

No images with same quality were found in both eyes. The image quality assessment was done only to pre-exclude images for which angle evaluation was not possible, which is not the main purpose of our study. Therefore, the further analysis and consideration are redundant and will blur the purpose and results.

3) A limitation of the GS-1, like AS-OCT is that it is performed in primary gaze without lens tilt, despite many clinical trials adding the dynamic component of gonioscopy (e.g. He et al 2019 Lancet) and clinical guidelines recommending identification of iridotrabecular contact that is only possible with lens tilt or off-axis gaze (e.g. Prum Jr et al 2016 Ophthalmology). This requires a comment on page 11.

Response

Thank you for the comment. We also agree with you as we described in paragraph 4 in our discussion. Moreover, according to your suggestion, we have changed the statement to make it easier for readers to understand in the manuscript L8-9 (pg 10).

4) The methods require a bit more transparency in the writing. Pages 12-13 for example appear particularly disorganized. The reader is introduced to five independent observers firstly on page 12, but then on page 13 line 3 onwards, a presumably singular observer is used for the intraobserver evaluation. However, then the authors state that a "second set" was used to further evaluate intraobserver agreements presumably for the five independent observers on line 16? Another example is page 13 line 13 - are the different randomized images from the same set or another prospectively collected set? The writing needs further clarification and would perhaps benefit from separation into subheadings that allow clarity in the methods.

Response 

Thank you for the comment. Following your advice, we have made the writing separated into subheadings and added the supplementary explanations in our manuscript. Please see page 11 to 12.

Results

1) There are a few issues regarding the presentation of Table 1. The authors did not seem to state the techniques used to measure many of the continuous variables reported in the table. Further to Table 1, why was pseudoexfoliative glaucoma and secondary open angle glaucoma separated? The reporting of the distribution of Shaffer's angle width but Scheie's angle pigmentation grade represents to the reader unnecessary differentiation and muddling of the grading schemes - why not just stick to one? Further, it could be argued that the reading of a mean and SD for these grades is questionable, due to their non-linearity and ordinal nature (see Phu et al 2020 OPO)... it may be more useful to break down the distribution of the exact numbers of each grade, especially since the Fleiss's kappa was used. Finally, the additive value of topical glaucoma medications is questionable: the paper is not really reporting on the contribution of these to angle grading and the hypothesis being tested is unlikely to be confounded by these factors. It would also be more informative to provide the proportion of poor image grades that were subsequently excluded from analysis in this table or in the results text. This would reinforce the authors' choice of a pragmatic approach and a more realistic impression of the deployment of GS-1.

Response

Thank you for the comment. Table 1 is only the demographics and clinical characteristics of the study subjects and stating all the measurement technique seems to be unnecessary and verbose. Regarding the second point, as we described in our introduction, pseudoexfoliation glaucoma is considered to be the most common type of secondary glaucoma and can advance rapidly with continuous high IOP and be refractory to several therapeutic interventions. Therefore, it is clinically important, and we analyzed pseudoexfoliation glaucoma separately. As for the third point, please see the above response for Methods 1). As for the fourth point, we disagree with you. The mean and SD have been used in the clinical gradings and was also used in some recent research for the angle evaluations with gonio-photos (see Teixeira et al., Eur J Ophthalmol. 2018, Matsuo et al, Br J Ophthalmol. 2019). Breaking down the distribution of the exact numbers of each grade would only make the manuscript unnecessarily redundant and obscure the purpose and results of this study. As for the final point, we can understand your meaning, however we cannot agree with you. This study already contains more than enough information to make a single treatise, and any further additions will not only unnecessarily obscure the meaning but will also lose sight of its original purpose. Again, Table 1 was just the demographics and clinical characteristics of the study subjects. Please understand the main focus of our research.

2) With regard to reporting Table 2 and the kappa values, it is unclear to the reader why Fleiss's kappa was used for binary variables (PAS and Sampaolesi line - present/absent). This confusion stems from the lack of contingency tables being presented in the results and the confusing methods. The authors should seek to clarify the most appropriate statistical method used. Fleiss's kappa may be appropriate for agreement between the observers but from the writing it is hard to say what was compared.

Response

We are sorry to mention that we disagree with you. Table 2 clearly shows the intraobserver reproducibility for angle evaluations between manual gonioscopy and automated gonioscope by one observer (MT). Because we cannot compare results analyzed by different statistical methods, we calculated both Fleiss' kappa coefficient and Kendall rank correlation coefficient for grading (nominal) and binary scales at the same time. Additionally, the Fleiss' kappa analysis in Table 2 is confirmation of the previous report (see Teixeira et al., Eur J Ophthalmol. 2018).

3) The reporting of closed angle detection is somewhat unclear and strikes the reader as somewhat misleading and requires clarification in writing. Not all of the 140 quadrants were closed, and so the question is whether the kappa was related to the binary outcome of "open" vs. "closed", or the agreement between observers (as noted above).

Response

Thank you for the query. As we described in the methods, we analyzed the intraobserver reproducibility for closed angle detection in each quadrant (n=140) and in each eye (n=35) comparing the outcomes with manual gonioscopy and those with automated gonioscope by one observer. Therefore, the kappa was related to the binary outcomes of "open" vs. "closed".

4) Page 17, line 14 refers to observers 1-5 and glaucoma specialists 1-3, but this was somewhat unclear from the methods as well.

Response

We described it in our manuscript. Please see L14-16 (pg 11). 

5) Table 3 (and similar): there was extreme variability in the kappa values across quadrants and across observers: could the authors comment on whether there was a systematic difference or random difference in the text? This is a very interesting result and the reader may be tempted to think it may be related to the distribution of angle grades, whereby some may be more obvious and easy to score (e.g. closed or very wide open angles) compared to the middling grades.

Response

Thank you for the comment. There would be the tendency that the intraobserver agreement values of the glaucoma specialist with enough experience using the GS-1 in the clinic were higher than those of the other ophthalmologists as described in the manuscript. On the other hand, the extreme variability in the kappa values across quadrants seems to be happened by chance. However, we cannot conclude it on a clear evidence because of the small sample size. Moreover, they deviate from the original purpose, thus we would like to leave them to the future large-scale research.

6) Overall, comments 3-5 above suggest that contingency tables may play a role in the reporting of the data

Response

We do not agree with you. Again, adding more data is redundant.

Discussion

1) The second paragraph of the discussion is confusing, and the reader is unsure what the authors are trying to say. The tone is confusing in whether 10% is a high number or a low number?

Response

Thank you for the advice. We have revised our manuscript to make it easier to understand for readers in the second paragraph.

2) With regard to objectivity, one questions the value of precise grading given the wealth of data required to change the management plan. For example, the decision to treat would be based on a multitude of factors aside from the angle appearance, including accessibility to health care, age, lens status, epidemiological risk factors and others (see Thomas and Walland 2013 CEO).

Response

We disagree with you. All the angle parameters in our study can be objectively evaluated based on the angle findings, and it is worthwhile to examine the accurate inter- and intra-observer agreements for angle evaluations only with the novel device in considering the reliability. Accurate angle assessments and clinical judgments based on it are separate issues, and how to apply them to each patient should be considered in the other future study.

3) Page 27, lines 6-11: with respect to the discussion on gonio photographs, there is a limitation previously noted in the literature that the authors seemed to have omitted: the fact that many of these are conducted in primary gaze, and lack the dynamism of manual gonioscopy, which is really the key advantage of the technique. This requires further discussion.

Response

Thank you for the comment. We described it in the manuscript L22-23 (pg 27), L1-3 (pg 28). Moreover, we have added the sentence to make it easier to understand in the discussion, L5-7 (pg 26).

4) A general question with regards to the comparison with the EyeCam, the light source for imaging and photography should be mentioned or discussed at some point. There are significant efforts taken in standardizing gonioscopy as a procedure to mitigate introduction of stray light through the pupil.

Response

Thank you for the query. According to your suggestion, we have added the related information in the manuscript, L12-16 (pg 27).

5) Another general comment with the discussion: there is far too much re-reporting of the results and this causes bloat in the discussion. The authors should revise this for succinctness.

Response

Thank you for the advice. We have revised and shortened the relevant part in our discussion.

6) The issue of angle grade distribution was mentioned on page 31, but the analysis seems to be lacking in terms of accounting for it as a confounding factor.

Response

We do not agree with you. We assessed the effects of image quality on observer agreements for angle evaluations using gonioscopic photos of GS-1. As the result, the observer agreements using grade 0 images were not always better than those using grade 1 images, probably because we completely excluded the grade 2 (blurred with no discernible details) images. Therefore, it is unlikely that the differences in image quality can be the confounding factor, and such detailed examinations that deviate from the original purpose are redundant and we would like to leave it to the other future study.

7) Another limitation of the grading system for image quality, which should also be noted in the references cited by the authors, is that the entirety of the image needs not be clear for evaluation. This does not seem to be captured with this method and should be discussed.

Response

We disagree with you. Same as on the previous reports, we assessed the image quality of the gonio-images and excluded the blurred images with no discernible details, which means all the unevaluable images was eliminated. You might be worried about the slightly vague but determinable images that were included in grade 1 by our definition, which we have already discussed in the above response to comment 6).

8) The authors raise an interesting point about the learning curve and the value of experience. This was discussed by Phu and colleagues and may be worthwhile integrated as part of this discussion (Phu et al 2019 OVS). What they may consider also acknowledging is the acquisition step. Though the processes described in the methods appear to be largely automated, there may be some component of interpreting whether the image is useable. A pragmatic approach as described by the authors would necessitate consideration of practical aspects such as the limitations and barriers to successful image acquisition and whether there are systematic errors in that process. A consecutive sampling strategy would be necessary as well as reporting of the errors associated with the technique. Finally, a calibration exercise would be suitable for this particular purpose.

Response

We disagree with your comments because they are out of our main focus of the study. The primary purpose was to analyze the reproducibility for the angle evaluations with the newly released gonioscopic camera, and it was the first verification to investigate to what extent glaucoma specialists and ophthalmologists made the same angle evaluations using the standardized gonio-images. As a result of the research, we could draw some important conclusions.

First of all, the recommended reference study by Phu et al was quite different from ours in the purpose, study design and methods. It was conducted with the medical records of gonioscopy results among different practitioners retrospectively, which must have full of bias including the observer bias to know the specific clinical information about the patients. Therefore, the result would be quite different from the pure angle evaluation and it is tough to draw the conclusion for the learning curve and the value of experience, and we would not like to make it reference.

With regard to the second point, as we described in the methods, we defined the exclusion criteria as the whole eyes with poor-quality images that had at least one grade 2 image in four ocular sectors, and finally 17 poor-quality eyes (23.9%) were excluded from angle evaluations. On the other hand, in the image acquisition with prototype GS-1, Teixeira et al reported that 22.7% eyes were excluded from the angle image grading as information would be lacking concerning at least two quadrants, and Shi et al demonstrated that 8.33% sections were not gradable owing to poor image quality. These studies’ exclusion criteria, conditions, and version of GS-1 were different from ours, thus we could not compare them directly, however, our exclusion rate was quite near to that of Teixeira’s. The reviewer seemed to stick to the further analysis for the limitations and barriers to successful image acquisition and whether there are systematic errors in the GS-1. However, the study already contains more than enough information to make a single manuscript, and the result is enough for our conclusions. A further detailed examination or analysis for the reviewer’s point is out of our main purpose and redundant.

Regarding the final points, the reviewer seemed to be misunderstanding that we did not conduct the consecutive sampling and calibration exercise, which were successfully done in the study. In fact, we did the consecutive sampling, therefore, we could demonstrate the successful image acquisition rate. On the other hand, the poor-quality GS-1 image eyes were excluded in the image selection process. Thus, strictly speaking, it was not intended for all consecutive patients who underwent GS-1, and we refrained from describing it. Moreover, we had several years of experience using GS-1 since it was prototyped (Matsuo et al., Br J Ophthalmol. 2019), so the calibration exercise in the image acquisition would be sufficient.

Minor comments

- Page 12, line 9-10: please clarify that the participants are the ophthalmologists

Response

Thank you for the comment. We have paraphrased the word for clarity in the manuscript L11-12 (pg 11).

- Page 13, line 22: nominal or ordinal?

Response

Nominal is correct. The analysis was performed by regarding the grading scale as the nominal scale to compare the results with each other.

- Table 1: Dioptor should be "Diopter"

Response

Thank you for making the typo. We have revised the word in Table 1.

- There is some inconsistency between "sectors" and "quadrants" used throughout the manuscript

Response

According to your advice, we have unified the word to sector.

- Page 25: there is some inconsistency in the way that the results are reported in the text. It is quite wordy and somewhat repetitive from the tables. The authors should note this trend throughout the manuscript and seek to clarify and write succinctly.

Response

The suggestion is same as the reviewer’s comment in Discussion 5). According to your advice, we have shortened and simplified the relevant part.

- Page 25, line 15: there should be a clearer distinction between an ophthalmologist (general?) and glaucoma specialist

Response

What part were you pointing out? We did not know the relevant part.

- Why were there "NA" results in Supp Table 1 - was not clear

Response

In our study, the analyses for Fleiss’ kappa statistic were performed in R statistical software version 3.5.3. and the calculations for Kendall rank correlation coefficients were conducted with JMP Pro 14 software. Because we are not the developer of those software, we do not know the exact answer. However, the discussion about it for S1 Fig is out of the main focus of the study and redundant.

- The units on Figures 1 and 2 are not clearly described in the Figure Captions. Although somewhat intuitive from the text, this may deserve further clarification.

Response

Thank you for your suggestion. We have added the supplementary explanation in the Figure legends.

---

## [Decision Letter · Decision Letter 1]

23 Feb 2021

PONE-D-20-24626R1

Intraobserver and interobserver agreement among anterior chamber angle evaluations using automated 360-degree gonio-photos

PLOS ONE

Dear Dr. Matsuo,

Thank you for submitting your manuscript to PLOS ONE. After careful consideration, we feel that it has merit but does not fully meet PLOS ONE’s publication criteria as it currently stands. Therefore, we invite you to submit a revised version of the manuscript that addresses the points raised during the review process.

We look forward to receiving your revised manuscript.

Kind regards,

Jinhai Huang, M.D.

Academic Editor

PLOS ONE

Journal Requirements:

Reviewers' comments:

Reviewer's Responses to Questions

**Comments to the Author**

1. If the authors have adequately addressed your comments raised in a previous round of review and you feel that this manuscript is now acceptable for publication, you may indicate that here to bypass the “Comments to the Author” section, enter your conflict of interest statement in the “Confidential to Editor” section, and submit your "Accept" recommendation.

Reviewer #1: (No Response)

2. Is the manuscript technically sound, and do the data support the conclusions?

Reviewer #1: Yes

3. Has the statistical analysis been performed appropriately and rigorously? 

Reviewer #1: Yes

4. Have the authors made all data underlying the findings in their manuscript fully available?

Reviewer #1: Yes

5. Is the manuscript presented in an intelligible fashion and written in standard English?

Reviewer #1: Yes

6. Review Comments to the Author

Reviewer #1: The authors have done an admirable job in addressing several of the comments and have engaged in a scholarly discussion. There are a few points of disagreement, but these are mostly related to the interpretation of the literature. I have listed several comments where there remain some disagreements, and I have some suggestions for the authors to improve clarity.

1) Authors' comment: However, we cannot agree with your idea that angle closure is important to identify as it has a different prognostic course compared to open angle glaucoma, because we sometimes experience cases of open-angle glaucoma (OAG) with end-stage visual field impairment at the first visit. If we could detect such patients early enough, we surely prevent the visual field loss due to OAG. Therefore, the early OAG detection and treatment is also clinically important.

Response: Whilst this is true (e.g. the work of Boodhna, Crabb and colleagues), the point of the paper does not appear to be related to POAG. The concept of making an introduction succinct is to engage with the reader in order to arrive at the core purpose of the study. I suggest that the authors use a new paragraph at page 4 line 20, beginning with the sentence "While" to disconnect POAG and PACG. On that point, whilst PLOS is a general science journal, the readership of a paper reporting on the outcomes of a highly specialised ophthalmic tool is most likely someone in the field.

Also, if the authors wish to describe the spectrum of angle closure disease at such length and comprehensiveness, it would be remiss of them to exclude the continuum between open and closed angles, as the prevention stage is touted by some to occur well before vision loss.

2) Authors' comment: We also believe the importance of configuration and distribution of pigment; however, it is impractical to evaluate the diagnostic reproducibility of all angle findings. Pigment dispersion syndrome is clinically rare, and the evaluation is out of our main purpose. Moreover, it is difficult to evaluate the observer agreement of rare finding with small sample size practically. Therefore, we limited the evaluation items in our study.

Response: There appears to be no statement or change made describing this limitation in the study - or if there is, it has not been clearly noted by the authors.

3) Authors' comment: Breaking down the distribution of the exact numbers of each grade would only make the manuscript unnecessarily redundant and obscure the purpose and results of this study. As for the final point, we can understand your meaning, however we cannot agree with you. This study already contains more than enough information to make a single treatise, and any further additions will not only unnecessarily obscure the meaning but will also lose sight of its original purpose. Again, Table 1 was just the demographics and clinical characteristics of the study subjects. Please understand the main focus of our research.

Response: This sounds almost contradictory. On one hand, the authors are willing to retain information on topical glaucoma medications even though its value has not been described anywhere in the text, nor is it immediately obvious to the reader - even a glaucoma expert - that its contribution is, even though the authors state it is a "main focus of their research". On the other hand, the contribution of angle grades on repeatability indices has more obvious value, but the authors have not provided justification on why it has been excluded.

4) Authors' comment: We are sorry to mention that we disagree with you. Table 2 clearly shows the intraobserver reproducibility for angle evaluations between manual gonioscopy and automated gonioscope by one observer (MT). Because we cannot compare results analyzed by different statistical methods, we calculated both Fleiss' kappa coefficient and Kendall rank correlation coefficient for grading (nominal) and binary scales at the same time. Additionally, the Fleiss' kappa analysis in Table 2 is confirmation of the previous report (see Teixeira et al., Eur J Ophthalmol. 2018).

Response: My point relates to comment 3 above - if you had the contingency table available, it would provide a more useful interpretation of repeatability at different angle grades. Kappa values are a useful statistical tool for comparison with previous studies - this is true - however if the authors are aiming for a pragmatic interpretation of the results, then kappa values become more subjective.

5) Authors' comment: We described it in our manuscript. Please see L14-16 (pg 11).

Response: this is written in an ambiguous manner, as it can be interpreted as 5 + 3 examiners, when it is 5, within which 3 were glaucoma specialists. This should be rephrased for clarity throughout the manuscript.

6) Authors' comment: Response

Thank you for the comment. There would be the tendency that the intraobserver agreement values of the glaucoma specialist with enough experience using the GS-1 in the clinic were higher than those of the other ophthalmologists as described in the manuscript. On the other hand, the extreme variability in the kappa values across quadrants seems to be happened by chance. However, we cannot conclude it on a clear evidence because of the small sample size. Moreover, they deviate from the original purpose, thus we would like to leave them to the future large-scale research.

and

We do not agree with you. Again, adding more data is redundant.

Response: Unfortunately, I disagree with these points. It is not redundant because it plans an important role in data visualisation and potentially to a less biased interpretation of the results.

7) Authors' comment: We disagree with you. All the angle parameters in our study can be objectively evaluated based on the angle findings, and it is worthwhile to examine the accurate inter- and intra-observer agreements for angle evaluations only with the novel device in considering the reliability. Accurate angle assessments and clinical judgments based on it are separate issues, and how to apply them to each patient should be considered in the other future study.

Response: Unfortunately, I also disagree with the conviction of the authors here and their statement that they are fully separate issues. It is important to remain skeptical of emphasising objective measures too much, and the authors have too quickly dismissed their contribution into a more holistic model of health care.

8) Authors' comment: Therefore, it is unlikely that the differences in image quality can be the confounding factor, and such detailed examinations that deviate from the original purpose are redundant and we would like to leave it to the other future study.

Response: I have noted that the authors appear to say a lot of things are redundant and deviate from the original purpose. I disagree with the notion that image quality - such an important component of any imaging based study - is redundant. As a reader, that is one of the first things one would question. The authors could consider at least reporting this in the supplementary material.

7. PLOS authors have the option to publish the peer review history of their article (what does this mean?). If published, this will include your full peer review and any attached files.

Reviewer #1: No

---

## [Author Response · Author response to Decision Letter 1]

6 Mar 2021

`````` March 3, 2020

Jinhai Huang, M.D.

Academic Editor

PLOS ONE

Dear Dr. Huang,

Ref: PONE-D-20-24626

Intraobserver and interobserver agreement among anterior chamber angle evaluations using automated 360-degree gonio-photos

PLOS ONE

We appreciate your reconsideration of our manuscript entitled “Intraobserver and interobserver agreement among anterior chamber angle evaluations using automated 360-degree gonio-photos” for publication in PLOS ONE as a Full Paper.

We received constructive feedback. We have revised our manuscript and made a point-by-point response to the comments as follows. We have worked hard to incorporate your feedback and hope that these revisions persuade you to accept our submission.

We state that this manuscript has not been published elsewhere and is not under consideration by another journal. All authors have approved the manuscript and agree with submission to PLOS ONE. Thank you!

Best regards,

Masato Matsuo, MD, PhD

Department of Ophthalmology,

Shimane University Faculty of Medicine,

Enya 89-1, Izumo, Shimane, JAPAN.

mmpeaceful@yahoo.ne.jp

matsuondmc@gmail.com

Journal Requirements:

Response

We have checked our reference list to ensure that it is complete and correct.

Reviewer #1’s Comments:

1) Reviewer‘s comment:

The first two paragraphs appear to be needlessly long: the authors could consider reshaping this to identify two succinct points: 1) angle closure as a disease entity is important to identify as it has a different prognostic course compared to open angle glaucoma; 2) secondary open angle (and closed angle) glaucomas require examination of the anterior chamber angle to identify underlying causes that may alter the treatment plan. Paragraph two in particular is a bit confusing in its tone with regard to the usefulness and limitations of gonioscopy. Moreover, there are some aspects of gonioscopy that are glossed over, such as the fact that grading systems for treatment titration rely solely on the gonioscopic impression (e.g. Prum Jr et al 2016 Ophthalmology), and that major clinical trials use gonioscopy as the technique of choice (He et al 2019 Lancet). Gonioscopy lenses such as the G6 in theory offer a more uninterrupted view of the anterior chamber angle so the statement on page 6, line 10 is not quite true.

Authors' comment:

Thank you for the comment, however we disagree with you. According to the journal’s submission guideline, we tried to provide the background that puts the manuscript into context and allows readers outside the field to understand the purpose and significance of the study. The readers are not limited to glaucoma specialists or ophthalmologists. Therefore, it is necessary to define the problem addressed and explain why it is important.

Regarding 1), we also think early detection of a narrow angle and peripheral anterior synechia (PAS) is vital to prevent primary angle-closure glaucoma (PACG) as described in paragraph 1. However, we cannot agree with your idea that angle closure is important to identify as it has a different prognostic course compared to open angle glaucoma, because we sometimes experience cases of open-angle glaucoma (OAG) with end-stage visual field impairment at the first visit. If we could detect such patients early enough, we surely prevent the visual field loss due to OAG. Therefore, the early OAG detection and treatment is also clinically important.

As for 2), we agree with your comment that secondary glaucomas require examination of the anterior chamber angle to identify underlying causes that may alter the treatment plan. Thus, we have added the statement in the manuscript L18-20 (pg 4). We believe that the change helps our manuscript more contextual. On the other hand, regarding the other indications, we do not agree with you. We pointed out the limitation of subjective assessments with gonioscopy in the manuscript L12-13 (pg 5). Additionally, we also know the major clinical trials use gonioscopy as the technique of choice, however the fact has nothing to do with the reliability of the test. Moreover, “gonioscopy can examine at one time only a limited contiguous portion of the iridocorneal angle” means that we can observe only a limited position of the angle with conventional gonioscopy at one time even if using G6 because of the manual nature. On the other hand, gonioscopic camera can take whole angle images simultaneously. Therefore, we pointed out the characteristics and limitations of the gonioscopy in the manuscript.

Reviewer’s comment: 

Whilst this is true (e.g. the work of Boodhna, Crabb and colleagues), the point of the paper does not appear to be related to POAG. The concept of making an introduction succinct is to engage with the reader in order to arrive at the core purpose of the study. I suggest that the authors use a new paragraph at page 4 line 20, beginning with the sentence "While" to disconnect POAG and PACG. On that point, whilst PLOS is a general science journal, the readership of a paper reporting on the outcomes of a highly specialised ophthalmic tool is most likely someone in the field.

Also, if the authors wish to describe the spectrum of angle closure disease at such length and comprehensiveness, it would be remiss of them to exclude the continuum between open and closed angles, as the prevention stage is touted by some to occur well before vision loss.

Response

Thank you for the comment. POAG is the most common type, and the condition is associated with an open anterior chamber angle without other known explanations (i.e., secondary glaucoma) for progressive glaucomatous optic nerve change. Thus, gonioscopic angle assessment is also essential for the management and our study is related to POAG. On the other hand, we have decided to add the supplementary explanations and divided into the paragraphs for easy understanding for the readers according to your suggestion in the manuscript L9-12 (pg 4), L1, L4-10 (pg 5).

2) Reviewer’s comment:

Aside from the pigmentation grade, the configuration and distribution of pigment is important clinically, such as in cases of burnt out pigment dispersion syndrome, mottled versus homogenous pigmentation - there are numerous citations for this, such as from Rob Ritch.

Authors' comment: 

We also believe the importance of configuration and distribution of pigment; however, it is impractical to evaluate the diagnostic reproducibility of all angle findings. Pigment dispersion syndrome is clinically rare, and the evaluation is out of our main purpose. Moreover, it is difficult to evaluate the observer agreement of rare finding with small sample size practically. Therefore, we limited the evaluation items in our study.

Reviewer’s comment: 

There appears to be no statement or change made describing this limitation in the study - or if there is, it has not been clearly noted by the authors.

Response

Thank you for the comment. We described the limitation in the manuscript L22-23 (pg 35), L1 (pg 36). Moreover, for better understanding, we have generated the additional ideas to supplement in the manuscript L2-3 (pg 36).

3) Reviewer’s comment:

There are a few issues regarding the presentation of Table 1. The authors did not seem to state the techniques used to measure many of the continuous variables reported in the table. Further to Table 1, why was pseudoexfoliative glaucoma and secondary open angle glaucoma separated? The reporting of the distribution of Shaffer's angle width but Scheie's angle pigmentation grade represents to the reader unnecessary differentiation and muddling of the grading schemes - why not just stick to one? Further, it could be argued that the reading of a mean and SD for these grades is questionable, due to their non-linearity and ordinal nature (see Phu et al 2020 OPO)... it may be more useful to break down the distribution of the exact numbers of each grade, especially since the Fleiss's kappa was used. Finally, the additive value of topical glaucoma medications is questionable: the paper is not really reporting on the contribution of these to angle grading and the hypothesis being tested is unlikely to be confounded by these factors. It would also be more informative to provide the proportion of poor image grades that were subsequently excluded from analysis in this table or in the results text. This would reinforce the authors' choice of a pragmatic approach and a more realistic impression of the deployment of GS-1.

Authors' comment:

Thank you for the comment. Table 1 is only the demographics and clinical characteristics of the study subjects and stating all the measurement technique seems to be unnecessary and verbose. Regarding the second point, as we described in our introduction, pseudoexfoliation glaucoma is considered to be the most common type of secondary glaucoma and can advance rapidly with continuous high IOP and be refractory to several therapeutic interventions. Therefore, it is clinically important, and we analyzed pseudoexfoliation glaucoma separately. As for the third point, please see the above response for Methods 1). As for the fourth point, we disagree with you. The mean and SD have been used in the clinical gradings and was also used in some recent research for the angle evaluations with gonio-photos (see Teixeira et al., Eur J Ophthalmol. 2018, Matsuo et al, Br J Ophthalmol. 2019). Breaking down the distribution of the exact numbers of each grade would only make the manuscript unnecessarily redundant and obscure the purpose and results of this study. As for the final point, we can understand your meaning, however we cannot agree with you. This study already contains more than enough information to make a single treatise, and any further additions will not only unnecessarily obscure the meaning but will also lose sight of its original purpose. Again, Table 1 was just the demographics and clinical characteristics of the study subjects. Please understand the main focus of our research.

Reviewer's comment:

This sounds almost contradictory. On one hand, the authors are willing to retain information on topical glaucoma medications even though its value has not been described anywhere in the text, nor is it immediately obvious to the reader - even a glaucoma expert - that its contribution is, even though the authors state it is a "main focus of their research". On the other hand, the contribution of angle grades on repeatability indices has more obvious value, but the authors have not provided justification on why it has been excluded.

Response

Thank you for the comment. It seems to be misunderstood. As we described in our previous response, information on topical glaucoma medications in Table 1 is only the demographics and clinical characteristics of the study subjects. We believe it is okay to keep it, however we would like to remove it if it interferes with the reader's understanding. Additionally, we found the typographical error in Table 1, and we have corrected it. Please see the underlined part. Moreover, according to your suggestion, we have added the contingency table demonstrating the distribution of angle grades in S1 Table. 

4) Reviewers' comment:

With regard to reporting Table 2 and the kappa values, it is unclear to the reader why Fleiss's kappa was used for binary variables (PAS and Sampaolesi line - present/absent). This confusion stems from the lack of contingency tables being presented in the results and the confusing methods. The authors should seek to clarify the most appropriate statistical method used. Fleiss's kappa may be appropriate for agreement between the observers but from the writing it is hard to say what was compared.

Authors' comment:

We are sorry to mention that we disagree with you. Table 2 clearly shows the intraobserver reproducibility for angle evaluations between manual gonioscopy and automated gonioscope by one observer (MT). Because we cannot compare results analyzed by different statistical methods, we calculated both Fleiss' kappa coefficient and Kendall rank correlation coefficient for grading (nominal) and binary scales at the same time. Additionally, the Fleiss' kappa analysis in Table 2 is confirmation of the previous report (see Teixeira et al., Eur J Ophthalmol. 2018).

Reviewers' comment: 

My point relates to comment 3 above - if you had the contingency table available, it would provide a more useful interpretation of repeatability at different angle grades. Kappa values are a useful statistical tool for comparison with previous studies - this is true - however if the authors are aiming for a pragmatic interpretation of the results, then kappa values become more subjective.

Response

Thank you for the suggestion. We have added the contingency tables in Supplemental material (S1 Table).

5) Reviewers' comment:

Page 17, line 14 refers to observers 1-5 and glaucoma specialists 1-3, but this was somewhat unclear from the methods as well.

Authors' comment:

We described it in our manuscript. Please see L14-16 (pg 11).

Reviewer’s comment: 

this is written in an ambiguous manner, as it can be interpreted as 5 + 3 examiners, when it is 5, within which 3 were glaucoma specialists. This should be rephrased for clarity throughout the manuscript.

Response

Thank you for the suggestion. As you pointed out, we have paraphrased the relevant part and added the supplemental explanation in the manuscript L18-19 (pg 7), L2 (pg 20), L2 (pg 21).

6) Reviewer’s comment:

Table 3 (and similar): there was extreme variability in the kappa values across quadrants and across observers: could the authors comment on whether there was a systematic difference or random difference in the text? This is a very interesting result and the reader may be tempted to think it may be related to the distribution of angle grades, whereby some may be more obvious and easy to score (e.g. closed or very wide open angles) compared to the middling grades.

and 

Overall, comments 3-5 above suggest that contingency tables may play a role in the reporting of the data

Authors' comment:

Thank you for the comment. There would be the tendency that the intraobserver agreement values of the glaucoma specialist with enough experience using the GS-1 in the clinic were higher than those of the other ophthalmologists as described in the manuscript. On the other hand, the extreme variability in the kappa values across quadrants seems to be happened by chance. However, we cannot conclude it on a clear evidence because of the small sample size. Moreover, they deviate from the original purpose, thus we would like to leave them to the future large-scale research.

and

We do not agree with you. Again, adding more data is redundant.

Reviewer’s comment: 

Unfortunately, I disagree with these points. It is not redundant because it plans an important role in data visualization and potentially to a less biased interpretation of the results.

Response

Thank you for the suggestion. We have added the contingency tables in Supplemental materials (S2 Table and S3 Table).

7) Reviewer’s comment:

With regard to objectivity, one questions the value of precise grading given the wealth of data required to change the management plan. For example, the decision to treat would be based on a multitude of factors aside from the angle appearance, including accessibility to health care, age, lens status, epidemiological risk factors and others (see Thomas and Walland 2013 CEO).

Authors' comment: 

We disagree with you. All the angle parameters in our study can be objectively evaluated based on the angle findings, and it is worthwhile to examine the accurate inter- and intra-observer agreements for angle evaluations only with the novel device in considering the reliability. Accurate angle assessments and clinical judgments based on it are separate issues, and how to apply them to each patient should be considered in the other future study.

Reviewer’s comment: 

Unfortunately, I also disagree with the conviction of the authors here and their statement that they are fully separate issues. It is important to remain skeptical of emphasising objective measures too much, and the authors have too quickly dismissed their contribution into a more holistic model of health care.

Response:

Thank you for the suggestion. We have added the contingency tables in Supplemental materials (S2 Table and S3 Table).

8) Reviewer’s comment:

The issue of angle grade distribution was mentioned on page 31, but the analysis seems to be lacking in terms of accounting for it as a confounding factor.

Authors' comment:

We do not agree with you. We assessed the effects of image quality on observer agreements for angle evaluations using gonioscopic photos of GS-1. As the result, the observer agreements using grade 0 images were not always better than those using grade 1 images, probably because we completely excluded the grade 2 (blurred with no discernible details) images. Therefore, it is unlikely that the differences in image quality can be the confounding factor, and such detailed examinations that deviate from the original purpose are redundant and we would like to leave it to the other future study.

Reviewer’s comment:

I have noted that the authors appear to say a lot of things are redundant and deviate from the original purpose. I disagree with the notion that image quality - such an important component of any imaging based study - is redundant. As a reader, that is one of the first things one would question. The authors could consider at least reporting this in the supplementary material.

Response

Thank you for the comment. Interpreting our results, we believe it is unlikely that the differences in image quality could be the confounding factor when grade 2 images were excluded. Moreover, as far as we know, there is no statistical method to adjust the confounding factors to evaluate the Fleiss’ kappa value.

---

## [Decision Letter · Decision Letter 2]

23 Apr 2021

Intraobserver and interobserver agreement among anterior chamber angle evaluations using automated 360-degree gonio-photos

PONE-D-20-24626R2

Dear Dr. Matsuo,

We’re pleased to inform you that your manuscript has been judged scientifically suitable for publication and will be formally accepted for publication once it meets all outstanding technical requirements.

Kind regards,

Jinhai Huang, M.D.

Academic Editor

PLOS ONE

Additional Editor Comments (optional):

Reviewers' comments:

Reviewer's Responses to Questions

**Comments to the Author**

1. If the authors have adequately addressed your comments raised in a previous round of review and you feel that this manuscript is now acceptable for publication, you may indicate that here to bypass the “Comments to the Author” section, enter your conflict of interest statement in the “Confidential to Editor” section, and submit your "Accept" recommendation.

Reviewer #1: All comments have been addressed

2. Is the manuscript technically sound, and do the data support the conclusions?

Reviewer #1: Yes

3. Has the statistical analysis been performed appropriately and rigorously? 

Reviewer #1: Yes

4. Have the authors made all data underlying the findings in their manuscript fully available?

Reviewer #1: Yes

5. Is the manuscript presented in an intelligible fashion and written in standard English?

Reviewer #1: Yes

6. Review Comments to the Author

Reviewer #1: (No Response)

7. PLOS authors have the option to publish the peer review history of their article (what does this mean?). If published, this will include your full peer review and any attached files.

Reviewer #1: No

---

## [Editor Report · Acceptance letter]

27 Apr 2021

PONE-D-20-24626R2 

Intraobserver and interobserver agreement among anterior chamber angle evaluations using automated 360-degree gonio-photos 

Dear Dr. Matsuo:

I'm pleased to inform you that your manuscript has been deemed suitable for publication in PLOS ONE. Congratulations! Your manuscript is now with our production department. 

Kind regards, 

on behalf of

Dr. Jinhai Huang 

Academic Editor

PLOS ONE